# MATFORMER: NESTED TRANSFORMER FOR ELASTIC INFERENCE

## ABSTRACT

Transformer models are deployed in a wide range of settings, from multi-accelerator clusters to standalone mobile phones. The diverse inference constraints in these scenarios necessitate practitioners to train foundation models such as PaLM 2, Llama, & ViTs as a series of models of varying sizes. Due to significant training costs, only a select few model sizes are trained and supported, limiting more fine-grained control over relevant tradeoffs, including latency, cost, and accuracy. This work introduces MatFormer[1], a nested Transformer architecture designed to offer elasticity in a variety of deployment constraints. Each Feed Forward Network (FFN) block of a MatFormer model is jointly optimized with a few nested smaller FFN blocks. This training procedure allows for the Mix'n'Match of model granularities across layers – i.e., a trained universal MatFormer model enables extraction of *hundreds* of accurate smaller models, which were never explicitly optimized. We empirically demonstrate MatFormer's effectiveness across different model classes (decoders & encoders), modalities (language & vision), and scales (up to 2.6B parameters). We find that a 2.6B decoder-only MatFormer language model (MatLM) allows us to extract smaller models spanning from 1.5B to 2.6B, each exhibiting comparable validation loss and one-shot downstream evaluations to their independently trained counterparts. Furthermore, we observe that smaller encoders extracted from a universal MatFormer-based ViT (MatViT) encoder preserve the metric-space structure for adaptive large-scale retrieval. Finally, we showcase that speculative decoding with the accurate and *consistent* submodels extracted from MatFormer can further reduce inference latency.

## 1 INTRODUCTION

Figure 1: MatFormer introduces nested structure into the Transformer's FFN block & jointly trains all the submodels, enabling free extraction of hundreds of accurate submodels for elastic inference.

Large Foundation models (Anil et al., 2023; OpenAI, 2023; Dehghani et al., 2023) are deployed in a variety of settings like real-time response on mobile phones or in batch setting on multi-cluster GPUs for web-scale serving. To handle such varied settings, each model family provides a few

---

[1]MatFormer stands for 🪆 **Mat**ryoshka Trans**former** due to the model's inherent nested nature.

*independently trained* models of different sizes. In order to cover a wide range of applications, typically these models' sizes are nearly linear on log-scale. For example, Llama family provides models with 7B, 13B, 33B and 65B parameters (Touvron et al., 2023a).

Such an approach has two key drawbacks: (a) as the models are independently trained, they incur significant overhead for colocation during inference and are not behaviorally consistent with each other which are detrimental to inference optimization techniques like speculative decoding (Leviathan et al., 2023) and model cascades (Wang et al., 2020c), and (b) due to training overhead, practitioners typically train only a few models which do not cover the entire set of downstream use-cases. For example, a deployment setup might, say, have the latency budget to support 40B parameter Llama model, but can only host a 33B variant because the next bigger model (65B) has significantly higher latency. So, one would need to settle for a less accurate model despite the larger latency budget. While model compression approaches aim to address this issue, they typically require additional training for each model that needs to be extracted. Furthermore, when applied to LLMs, these techniques are known to significantly drop the accuracy (Jaiswal et al., 2023).

In this paper, we propose MatFormer, a natively elastic Transformer (Vaswani et al., 2023) architecture that allows for training one *universal* model which can be used to extract hundreds of smaller submodels without *any additional training* (Figure 1). MatFormer is a general architecture that can be applied to both encoders and decoders, is domain agnostic, and is compatible with most design choices and training pipelines of large Transformer-based models – LLMs & ViTs.

MatFormer follows the principle of matryoshka representation learning (Kusupati et al., 2022), to introduce nested substructure inside the standard Transformer block. Formally, MatFormer defines a Transformer blocks $T_i$, such that, $T_1 \subset T_2 \subset \cdots \subset T_g$, where $g$ is the number of nested transformer blocks, and $T_i \subset T_{i+1}$ relation indicates that the parameters of $T_i$ are contained in those of $T_{i+1}$. MatFormer can induce such sub-structure in both the attention and the feedforward network (FFN) blocks of the Transformer (see Figure 1). Consider a FFN block that has $d_{\text{ff}}$ neurons in the hidden layer. Then, MatFormer induces matryoshka structure on these neurons, where $T_i$ contains the first $m_i$ neurons and $1 \leq m_1 \leq m_2 \cdots \leq m_g = d_{\text{ff}}$ represent the number of neurons for each granularity or sub-model. Intuitively, this implies that the first $m_1$ neurons are "most significant" neurons as they belong to all the blocks followed by the next $m_2 - m_1$, and so on. We can form a similar substructure on the attention heads, with the heads being organized from "most" to "least" significant, where the more significant heads are shared by more sub-models. That is, we use only the first $m_i$ attention heads for the $i$th granularity. In fact, we can also introduce this sub-structure in the token embedding ($d_{\text{model}}$) supplied to each Transformer block.

However, in most LLMs and ViTs, the FFN block in the Transformer accounts for more than $60\%$ non-embedding parameters and is responsible for the largest chunk of latency during inference. So, in this work, we focus on inducing the MatFormer's nested sub-structure in the FFN block. We then stack the individual blocks (for $l$ layers) to form $g$ nested models ($\mathcal{M}_{1 \cdots g}$) with shared parameters i.e., $\mathcal{M}_i \subset \mathcal{M}_{i+1}$. Finally, we jointly train these $g$ models by combining each model's loss.

This leads to a natural question: can one extract more than $g$ models after inducing the MatFormer structure? Yes, in fact, it is possible to extract exponentially many models. Using the trained MatFormer blocks $T_1, \ldots, T_g$ at each layer, one can form new models by Mix'n'Match, i.e., by taking an arbitrary combination of these blocks across layers. For example, in the first layer, one can select $T_g$, the largest block, choose $T_2$ in the second layer, and so on, forming $g^l$ different models. As we explicitly optimized only for $g$ models, instead of the exponentially many models, are the extracted models accurate? Surprisingly, in multiple settings, and for a various model sizes, we observe that the extracted models indeed are accurate, with accuracy scaling with the size of the extracted model.

We train Matformer-based decoder-only Language Models (MatLM) up to 2.6B parameters and observe that: (a) MatLMs explicitly trained with $g$ exponentially spaced granularities almost match validation loss and one-shot downstream evals of respective $g$ baseline models trained independently from scratch, (b) our extracted models using Mix'n'Match lie on the accuracy-vs-parameters trade-off curve generated by the $g$ explicitly trained models, (c) through scaling experiments we observe that the loss vs compute law for different MatFormer models remains similar to vanilla Transformer models across different granularities and (d) the submodels extracted from MatLM have highly consistent behavior that is highly desirable for inference optimizations and deployment across scales.

We further studied MatFormer-based ViT models (MatViT) and have similar observations as MatLM. For example, MatViT-L/16 improves the accuracy of the standard ViT-L/16 model on ImageNet-1K, and the extracted sub-models all match or even perform better than the independently trained baselines. Furthermore, we demonstrate that, due to high consistency, MatViT models can be used as "elastic encoders" for adaptive image retrieval. That is, the metric-space of an image encoded by the universal (i.e. the largest) MatViT model is roughly preserved by the nested submodels. Hence, based on query complexity, system load, and various other considerations, we can use one of the extracted MatViT encoders at inference time for retrieval on a fixed corpus encoded by the universal model – providing over $40\%$ lesser compute overhead with $< 0.5\%$ drop in accuracy.

**We make these key contributions:**

1. We introduce MatFormer, which incorporates a nested sub-structure within the standard Transformer and jointly optimizes all the $g$ granularities to produce a single, universal elastic model.
2. Employing Mix'n'Match of granularities across layers in a universal MatFormer model yields hundreds of accurate and consistent submodels without any additional training cost (Section 3).
3. MatFormer generalizes effectively to both decoder-only language models (MatLM) and vision encoders (MatViT), scaling as reliably and accurately as the standard Transformer, while enabling significantly faster autoregressive generation and large-scale adaptive dense retrieval (Section 4).

## 2 RELATED WORK

A standard Transformer (Vaswani et al., 2023) has become the unifying model architecture for foundation models (Bommasani et al., 2021) across modalities like language (Brown et al., 2020), vision (Dehghani et al., 2023) and audio (Radford et al., 2023). While extremely powerful, the standard Transformer block is not natively elastic in a way that enables large-scale adaptive and flexible deployment across various resource constraints. To cater to the plethora of deployment requirements, existing solutions include training a family of models of varying sizes (Anil et al., 2023; Touvron et al., 2023b), post-hoc efficiency techniques like quantization (Dettmers & Zettlemoyer, 2023), pruning (Lagunas et al., 2021), distillation (Sanh et al., 2019) and mixture of varying capacity experts (MoE) (Zhang & Ma, 2012). However, these solutions often are specific to the single constraint at hand, and require additional training or trade-off memory/compute during inference making them far from being a truly elastic solution for adaptive deployment. Lastly, Transformer based LLMs are often sped-up during inference with techniques like speculative decoding (Leviathan et al., 2023; Chen et al., 2023) – that benefits from the smaller draft & the larger verifier models having similar behavior – or early exiting (Schuster et al., 2022) to enable real-time deployment.

Obtaining multiple smaller models from a single model has been explored in the past (Yu et al., 2018; Yu & Huang, 2019; Cai et al., 2019; Grimaldi et al., 2022; Cai et al., 2021) with most works focusing on CNN encoders. Specifically, OFA (Cai et al., 2019) creates a universal CNN model which is used to extract and finetune submodels for a handful of deployment constraints while slimmable networks (Yu et al., 2018) optimize for limited preset widths and require explicit training to interpolate for a few more intermediate widths (Yu & Huang, 2019). NAS techniques that sample random (not nested) subnetworks during training at each step, and then find the subnetwork architecture to retrain from scratch before deployment have been explored (Wang et al., 2020b). These techniques fall short of being truly elastic and come with significant training overheads. More recently some of them have been extended to Transformer encoders (Chavan et al., 2022; Hou et al., 2020; Salehi et al., 2023) for extracting sub-models in both static or dynamic settings but fail at extending further to decoder-only language models. While not in the weight space, matryoshka representation learning (Kusupati et al., 2022) & FlexiViT (Beyer et al., 2023) showcase elasticity in output & input spaces respectively by smoothly spanning deployment constraints with minimal overhead. MatFormer, in contrast, builds upon these works by nested the weight space instead to enable truly elastic and adaptive Transformer-based (decoder & encoder) models that span all the accuracy-vs-compute tradeoff (statically or dynamically) with minimal changes and training overhead (Figure 1). Finally, we also point the readers to SortedNet (Valipour et al., 2023), a concurrent work with similar goals applied to encoders, which optimizes many sampled submodels (akin to prior works) unlike MatFormer's joint optimization of a few (typically 4) nested submodels.

## 3 MATFORMER

In this section, we define MatFormer's nested substructure (Section 3.1) and discuss its training procedure for a chosen $g$ model granularities (Section 3.2). We then discuss elastic inference using Mix'n'Match models (Section 3.3) from MatFormer along with its deployment considerations.

### 3.1 MATFORMER STRUCTURE

MatFormer defines $g$ Transformer blocks $T_i$, such that, $T_1 \subset T_2 \subset \cdots \subset T_g$ where $T_i \subset T_{i+1}$ indicates that the parameters of $T_i$ are contained in those of $T_{i+1}$. While it is possible to impose such a structure on any part of the Transformer, we select the FFN block to define our method and present our experiments, as the model size and computational cost of a Transformer is dominated (around $60\%$ for LLMs and ViTs) by the FFN block (see Appendix B).

The Transformer FFN block has a single hidden layer with $d_{\text{ff}}$ neurons and both input and outputs in $\mathbb{R}^{d_{\text{model}}}$, and fixed FFN ratio $:= d_{\text{ff}}/d_{\text{model}}$ (typically $\geq 4$). MatFormer introduces the matryoshka nested structure with $g$ granularities on the hidden representation of the FFN block. Concretely, a nested sub-block of the Transformer, $T_i$ contains the first $m_i$ neurons of the FFN and $1 \leq m_1 \leq \cdots \leq m_g = d_{\text{ff}}$ represent the number of neurons for each granularity or sub-model. So, depending on the chosen granularity the FFN operation of $T_i$ i.e., $T_i^{\text{FFN}}$ on an input $x \in \mathbb{R}^{d_{\text{model}}}$ is:

$$T_i^{\text{FFN}}(x) = \sigma(x \cdot \mathbf{W}_1[0:m_i]^\top) \cdot \mathbf{W}_2[0:m_i], \tag{1}$$

where the weight matrices of FFN are $\mathbf{W}_1, \mathbf{W}_2 \in \mathbb{R}^{d_{\text{ff}} \times d_{\text{model}}}$ and bias terms are omitted for simplicity. $\mathbf{W}_1[0:k]$ denotes the submatrix with the first $k$ rows of $\mathbf{W}_1$. Finally, $\sigma$ is a non-linearity often set to GELU (Hendrycks & Gimpel, 2016) or squared ReLU (So et al., 2021). In this work, we chose the $g = 4$ exponentially spaced granularities with FFN ratios of $\{0.5, 1, 2, 4\}$ i.e., the nested hidden neurons are of the sizes $\{\frac{d_{ff}}{8}, \frac{d_{ff}}{4}, \frac{d_{ff}}{2}, d_{ff}\}$.

With the nested MatFormer blocks $T_1, T_2 \ldots T_g$, we can combine these to form a MatFormer model, with $g$ nested submodels $\mathcal{M}_1 \subset \mathcal{M}_2 \ldots, \subset \mathcal{M}_g$ where $\mathcal{M}_i \leftarrow [T_i]^{\times l}$, i.e., $\mathcal{M}_i$ is formed by stacking $T_i$ for $l$ layers. The input and output embedding matrices are shared across the models.

### 3.2 TRAINING

For a Transformer model $\mathcal{M}$, the forward pass on an input $x$ is denoted by $\mathcal{M}(x)$ and let $\mathcal{L}$ denote the loss function between the output and the target $y$: $\mathcal{L}(\mathcal{M}(x), y)$.

MatFormer relies on a simple training strategy of jointly optimizing all the $g$ nested submodels together. To this end, we set the MatFormer loss as a weighted average of loss of $g$ submodels and train for it using the standard stochastic gradient-based optimizers (Shazeer & Stern, 2018):

$$\mathcal{L}_{\text{JOINT}}(x, y) = \sum_{i=1}^{g} \lambda_i \cdot \mathcal{L}(\mathcal{M}_i(x), y), \tag{2}$$

where $\lambda_i > 0$ is the weight of $i$-th granular submodel. In this paper, we set $\{\lambda_i\}_{i=1\ldots g}$ to be uniform i.e., $1/g$ but explore tuning $\{\lambda_i\}_{i=1\ldots g}$ in Appendix E.4 to further improve MatFormer.

The joint training in MatFormer involves one forward pass per each of the $g$ submodels and benefits from portions of shared computation during backpropagation. MatFormer training results in $g$ accurate nested submodels $\mathcal{M}_{1\ldots g}$ inside the universal MatFormer model ($\mathcal{M}_g$). Note that this simple strategy outperforms various other training techniques (Appendix E.2). Finally, instead of pretraining models with MatFomer structure, we can also induce this structure via finetuning.

MatFormer training is $\sim 15\%$ faster (for $g = 4$) than training all the Transformer based equivalent submodels independently (Appendix B). However, MatFormer also enables the extraction of hundreds of smaller submodels along the accuracy-vs-compute curve traced by the $g$ explicitly optimized submodels (Section 3.3). These models emerge for free using Mix'n'Match during inference and drastically reduce the amortized training cost per model obtained through MatFormer. The joint optimization, even without self-distillation from $\mathcal{M}_g$, results in smaller submodels that have highly consistent behavior (Section 3.4) with the universal model. Finally, in Appendix B.1, we argue that the training efficiency of MatFormer can be significantly improved through various optimizations.

### 3.3 MIX'N'MATCH

At inference time, it is trivial to extract one of the $g$ submodels $\mathcal{M}_1 \subset \mathcal{M}_2 \ldots, \subset \mathcal{M}_g$ by stacking the corresponding Transformer block $T_i$ across layers. However, by selecting different granularity for each MatFormer layer, it is possible to generate a combinatorially large number of accurate smaller models for free. We call this simple procedure *Mix'n'Match* and observe that these additional model granularities –which were never explicitly optimized – are highly performant.

In fact, we can further increase the number of extracted models by generating interpolating blocks between fixed granulaties (Kusupati et al., 2022). For example, we can generate a $\widetilde{T}$ block that uses first $\frac{1}{2}(m_i + m_{i+1})$ neurons in the FFN layer which still tends to be highly accurate.

To summarize, given a computational budget, we can extract a highly accurate model with Mix'n'Match for the constraints rather than using a smaller less accurate model or training a model for this specific constraint (Sections 4.1.1 & 4.2). We note that a compute constraint can be satisfied by various Mix'n'Match models with different accuracies, making identifying the best Mix'n'Match configurations without downstream validation is an exciting direction for future work.

### 3.4 DEPLOYMENT

During deployment, all we need to store is the single universal MatFormer model for different types of elastic inference depending on the constraints. In the case of static workloads, where compute resources are known beforehand and the inputs remain relatively similar in difficulty, one can choose the most accurate static submodel for the constraints using Mix'n'Match. This eliminates the usage of a less accurate preexisting model or training of a new one for the specific constraints.

For dynamic workloads, where the compute resources or the input hardness change on the fly, we can use the universal MatFormer model to dynamically extract the optimal submodel for token-based routing in LLMs akin to MoE (Kudugunta et al., 2021; Li et al., 2022) and elastic encoders in dense retrieval (Section 4.2.2). This works largely because all the extracted submodels have high behavioral *consistency* with universal MatFormer model (Section 4.1) – minimizing the drift across predictions from various submodels. We measure the consistency between two generative models as the *percentage of matching tokens* generated by them for the same prefix or using the *KL divergence* of the smaller model outputs with the larger model outputs – this accounts for potential sampling strategies in decoding. This highly consistent nature of MatFormer results in superior inference time speedups for techniques like speculative decoding (Leviathan et al., 2023) (Section 4.1.1) and can assist in reducing prediction drift between cross platform deployments. We also show that higher model consistency also aids metric-space structure preservation in encoder models (Section 4.2.2).

## 4 EXPERIMENTS

In this section, we empirically evaluate MatFormer across modalities (language and vision), model classes (decoder and encoder), and scales (up to 2.6B parameters). Specifically, we train and analyze MatFormer-based decoder-only Language Models – MatLMs (Section 4.1) – and encoder-only Vision Transformers – MatViT (Section 4.2) models with $g = 4$ nested granularities across various model sizes. For a fair comparison, we also independently train the Transformer baseline for the submodel of each granularity across model sizes for the same tasks. We primarily focus on the elastic deployment of MatFormer-based models (Sections 4.1.1 & 4.2) for tasks spanning from one-shot generative evals to adaptive image retrieval. Additionally, we also investigate the reliable scaling behavior (Kaplan et al., 2020) of the MatFormer models (Section 4.1.2).

### 4.1 MATLM: MATFORMER LANGUAGE MODELS

We build MatFormer-based decoder-only Language Models – MatLMs – and contrast them to their vanilla Transformer counterparts (LMs) (Liu et al., 2018). The LMs broadly follow the training pipeline and procedure outlined by Thoppilan et al. (2022). For each MatLM model with a set $d_{\text{model}}$, we jointly optimize for $g = 4$ nested granularities represented by FFN ratios of $\{0.5, 1, 2, 4\}$ – i.e., only the hidden representation size of the FFN block changes. We denote these submodels as MatLM – {S, M, L, XL} in increasing order of model size and refer to MatLM-XL as the universal

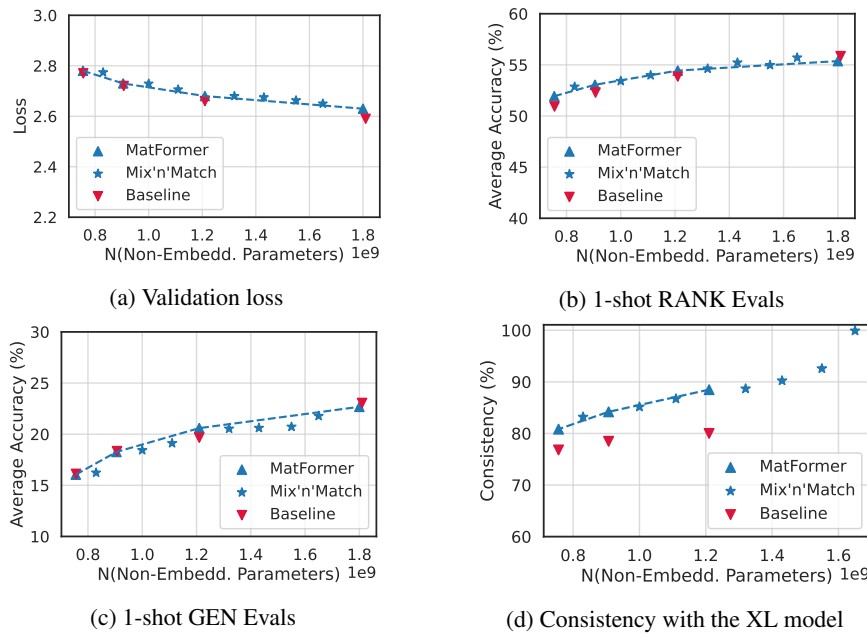

Figure 2: Validation loss & one-shot downstream evaluation scores for the 2.6B MatLM & baseline models. Mix'n'Match helps generate accurate and more consistent models from MatLM that lie on the performance-vs-compute curve spanned by the explicitly optimized submodels.

MatLM. For baselines, we train vanilla Transformer models with comparable architectures. That is, for each MatLM, we train 4 separate baseline models with FFN ratios of $\{0.5, 1, 2, 4\}$ for a fixed $d_{\text{model}}$ denoted as Baseline – {S, M, L, XL}. We evaluate these models on validation loss (= log perplexity) and average accuracy on 26 English tasks similar to (Brown et al., 2020; Du et al., 2022; Anil et al., 2023). Of these 26 tasks, we group 5 tasks that require generating multiple tokens under "GEN" and the remaining tasks that involve choosing an option from the input text under "RANK". Please see Appendix A for further details on training, evaluation, and the datasets.

### 4.1.1 ELASTIC INFERENCE WITH MATLM

To showcase elastic inference, we evaluate the 2.6B parameter MatLM models on its ability (a) to provide models spanning the accuracy-vs-compute curve using Mix'n'Match (Section 3.3) and (b) to improve post-hoc inference optimization techniques like Speculative Decoding (Leviathan et al., 2023) to further speed-up accurate auto-regressive generation.

**Accurate MatLM submodels for every constraint for free with Mix'n'Match.** Leveraging Mix'n'Match, a MatLM can provide accurate models for every compute constraint (between S and XL), not just the explicitly optimized granularities {S, M, L, XL}. We evaluate the impact of Mix'n'Match on the 2.6B parameter MatLM in Figure 2 through validation loss and downstream evals and contrast them to four granularities {S, M, L, XL} of the 2.6B baseline LM (all trained independently). In Figures 2a, 2b & 2c, we show that all MatLM – {S, M, L, XL} models all perform as well as their corresponding baselines – with marginal improvements and drops across the scale.

In Figure 2a we see that Mix'n'Match helps obtain many models on the optimal loss-vs-compute curve at zero cost. Moreover, downstream eval tasks on these Mix'n'Match models also mimic this trend, as shown in Figures 2c & 2b. In a deployment setting that only has $55\%$ of the required compute resources needed for the MatLM-XL model, it is now possible to have a Mix'n'Match submodel with $< 2\%$ accuracy drop on RANK evals. Without elastic deployment due to Mix'n'Match, we would see a $> 2.5\%$ accuracy drop due to the use of the MatLM-M model. Note that we highlight only a few of the hundreds of accurate Mix'n'Match models along the curves. We discuss additional details and results on the Mix'n'Match procedure in Appendix C.

**MatLM submodels speed up speculative decoding.** Speculative decoding leverages an accurate lightweight LM as a draft model to autoregressively generate a few tokens, followed by verifying these drafts with a larger model through parallel decoding on the generated tokens. When the draft

is inaccurate, the draft model is rolled back and reset to the larger model's output. This results in considerable inference speed-up for the *same accuracy as the large model*. We point the reader to the original paper for a more detailed explanation (Leviathan et al., 2023).

Slow down of this algorithm stems from cases where the smaller model's predictions disagree with the larger model. A draft model that is significantly more consistent with the larger verifier model would lead to less rollbacks of the draft predictions and therefore lower latency. As seen in Figure 2d the MatLM submodels can be up to 8.5% more consistent than the baselines to their corresponding XL model. The significant gap persists even in the KL divergence variant of consistency with the XL model's outputs (see Figure 8 in Appendix). This improved consistency along with the need for only a single universal model positions MatLM favorably to improve techniques that require draft and verifier models such as speculative decoding.

Table 1: Inference time speed-ups over a standard 2.6B model through speculative decoding using a 1.5B (S) draft and 2.6B (XL) verifier model.

| Speculative Decoding | LAMBADA | TriviaQA |
|---|---|---|
| Baseline | 1.10× | 1.08× |
| MatLM | 1.14× | 1.11× |
| + shared attention cache | 1.16× | 1.14× |

Table 1 shows the inference time speed-ups from speculative decoding using the S and XL submodels of the 2.6B language model for drafting and verification respectively. Speculative decoding with independently trained baseline LMs results in a speed-up of up to 10% over the standard autoregressive decoding of the 2.6B-XL model. But MatLM-based speculative decoding is up to 6% faster than traditional speculative decoding. This additional speed-up can be primarily attributed to the more consistent nature of MatLM-based drafter and verifier models and is further boosted by the ability to share attention cache across models from MatLM which is infeasible for the baselines (see Appendix B.2). Finally, MatLM further reduces the memory overhead for inference by removing the need to have two models during resource-constrained deployment.

### 4.1.2 MATLM SCALES AS WELL AS VANILLA TRANSFORMER LMS

Now that we have established that a 2.6B MatLM model and its submodels are as accurate as the baseline Transformer LMs, we want to examine the scalability of training MatLM models. So, we study the scaling properties (Kaplan et al., 2020; Hoffmann et al., 2022) of MatLMs and compare them to vanilla Transformer baseline LMs trained for the same number of tokens. We train models

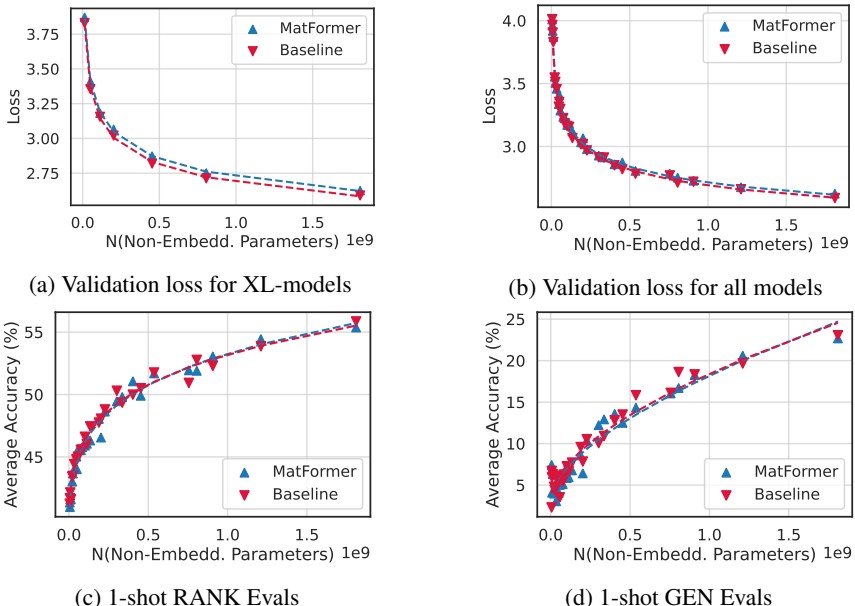

(a) Validation loss for XL-models

(b) Validation loss for all models

(c) 1-shot RANK Evals

(d) 1-shot GEN Evals

Figure 3: We train various decoder-only MatLM models at a range of sizes from 78M to 2.6B parameters and observe the scaling trends of all granularities (S, M, L, XL) for validation loss and 1-shot downstream evaluation scores. We find that the MatLM-XL models across scales mimic the training trends of Baseline-XL models. Interestingly, we also note that that validation loss and downstream evaluations follow the *scaling trends of the XL-models across all granularities*.

ranging from 78M to 2.6B parameters on 10B to 160B tokens and plot the validation loss for MatLM – {S, M, L, XL} compared against their baselines in Figure 9.

First, in Figure 3a, we observe that the training of MatLM-XL models across model sizes scale as reliably as the Baseline-XL LMs for loss vs. number of parameters. However, Figure 3b interestingly shows that it is not just the XL models but rather all the nested submodels, irrespective of granularity {S, M, L, XL}, of MatLM and Baseline that follow the same scaling trend. Therefore, we fit a scaling law according to the number of non-embedding parameters ($N$) and training tokens ($D$) for all possible submodels for both MatLMs and the baselines in Table 2. We observe that the fitted parameters are extremely similar, suggesting that MatLMs scale similarly to vanilla Transformer LMs.

In Figures 3c & 3d we also find that the downstream evals for MatLM are within $0.5\%$ of the baselines, with the smaller submodels even outperforming the baselines at scale. Finally, Figure 9f in the Appendix shows that the MatLM submodels are more consistent with their XL model compared to the baseline counterparts across scales.

Table 2: Fitted parameters for the scaling equation: $\text{Loss}(N, D) = a \cdot (ND)^b + c$

|  | a | b | c |
|---|---|---|---|
| Baseline | 20.917 | -0.119 | 1.868 |
| Matformer | 17.516 | -0.114 | 1.845 |

We note that the scaling law equation does not capture how (1) MatLMs have been optimized for multiple submodels and even have performant submodels that have not been explicitly optimized for (Section 4.1.1), and (2) MatLMs and baselines of the same size have different training FLOPs per step. We leave formulations that capture these subtleties to future work and further discuss this in Appendix D.1. We provide full results split by granularity in Appendix D.

### 4.2 MATViT: MATFORMER VISION TRANSFORMERS

In this section, we extend MatFormer to Vision Transformer (ViT) (Dosovitskiy et al., 2020) based computer vision encoder models. MatFormer-based ViT – MatViT – enables elastic inference for fundamental tasks like image classification and retrieval. To this end, we train the MatFormer variant of the standard ViT-B/16 and ViT-L/16 models – MatViT-B/16 and MatViT-L/16 that are trained with $g = 4$ prechosen nested granularities (FFN ratios of $\{0.5, 1, 2, 4\}$). B/16 models are trained on ImageNet-1K (Russakovsky et al., 2015) with AugReg (Steiner et al., 2021) while L/16 models are pretrained on ImageNet-21K (Deng et al., 2009) followed by finetuning on ImageNet-1K. All models are trained with the training setup and optimal hyperparameters of the standard ViT variants from the Scenic library (Dehghani et al., 2022).

#### 4.2.1 IMAGE CLASSIFICATION

For image classification, we evaluate both ViT & MatViT models on ImageNet-1K. Figure 4a shows that the explicitly optimized granularities in MatViT result in as accurate models as the independently trained baselines for the B/16. However for L/16, as shown in Figure 4b, we see that the MatViT models are up to $0.35\%$ more accurate than the baseline for the same inference cost.

We then explore using MatFormer at different training stages with a $2 \times 2$ grid of pretraining-finetuning pairs (Table 7 in Appendix F.1) and find that using a MatFormer during pretraining helps

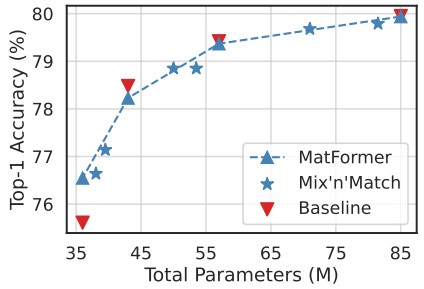

(a) B/16 trained on ImageNet-1K with AugReg

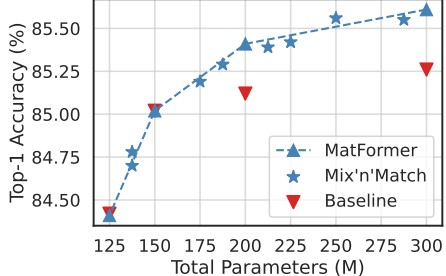

(b) L/16 pretrained on IN-21K → ImageNet-1K.

Figure 4: MatViT variants match or outperform standard ViT models on ImageNet-1K classification and provide free extracted models that span the accuracy-compute curve through Mix'n'Match.

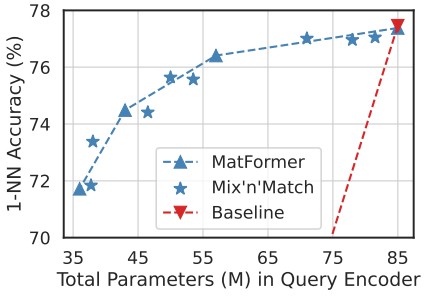 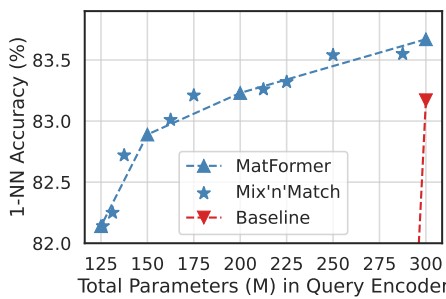

(a) B/16 trained on ImageNet-1K with AugReg      (b) L/16 pretrained on IN-21K → ImageNet-1K.

Figure 5: MatViT natively enables elastic encoders for adaptive retrieval that can be used for real-time query side computation while retaining strong accuracy on ImageNet-1K, unlike the baselines.

bring more accurate and flexible encoders for downstream use. Further, finetuning using MatFormer enhances elastic deployment depending on the constraints at hand through Mix'n'Match.

**Adaptive Encoders with Mix'n'Match.** Furthermore, our Mix'n'match models' accuracy almost lies on the line joining accuracy of explicitly trained granularities. In scenarios where, say, an application can host 50M parameter B/16 model, MatViT can provide $0.8\%$ more accurate model than the current approach which would host the largest baseline model with $\leq$ 50M parameters.

During deployment, the universal MatViT model can be stored in memory and depending on the compute constraints be used to extract an adaptable smaller model to maximize accuracy with the available resources at that moment. Currently, we find the Mix'n'Match models on the accuracy-compute curve through a quick inference on the validation set. While relatively scalable, this points to the need for optimal budget allocation across layers in neural networks (Kusupati et al., 2020).

### 4.2.2 ADAPTIVE IMAGE RETRIEVAL

The goal of image retrieval is to find semantically similar images – e.g. images from the same class – using representations obtained from a pretrained encoder (Chen et al., 2022). Standard approach is to encode the database images as well as query image with same encoder and run nearest neighbor retrieval for the query embedding. While we can embed database images with an expensive encoder, the query encoder generally has to be real-time. Furthermore, the setting of query encoding might be varied, e.g., on-device vs. cloud processing, varying query load and query complexity. Current solutions have to stick to a fixed encoder thus compromising on accuracy or cost for various settings.

Given the elastic nature of MatViT, it is a good candidate for query encoder. However, retrieval also requires that submodels preserve distances between fixed database (with large encoder) and query embeddings across all the granularities. If we use smaller baseline ViT models only for query encoding, these distances are not preserved and lead to nearly 0 retrieval accuracy (see Figure 5).

We evaluate both ViT and MatViT encoders on ImageNet-1K for image retrieval. We compute 1-nearest neighbor (NN) accuracy using the representation vector of the [CLS] token (also see Appendix F.2). Figure 5 shows that submodels extracted from MatViT can approximately preserve distances and provide significantly more flexibility. For example, with a loss of $< 0.5\%$ accuracy, MatViT-L/16 can reduce compute cost by $40\%$. To our knowledge, this is the first result of its kind and opens up a wide variety of adaptive inference strategies for large-scale semantic search.

## 5 CONCLUSIONS

In this work we presented MatFormer, a natively elastic Transformer architecture that allows training a single universal model which can be used to extract hundreds of smaller accurate submodels at zero additional cost at deployment time. We find that the MatFormer Language Model (MatLM) matches the perplexity & 1-shot accuracy of independently trained models. In fact, MatLM demonstrates an interesting loss-vs-compute scaling curve that is nearly *independent* of trained granularity indicating robust generalization to *extremely* large models as well. Finally, MatFormer submodels enable diverse inference time speedups like faster autoregressive generation with speculative decoding and elastic query encoders for adaptive dense retrieval across modalities.

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
