# A    IMPLEMENTATION DETAILS

## A.1    ARCHITECTURE AND TRAINING

For our experiments, we train a range of MatLMs varying from size 78M to 2.6B for 10B-160B tokens – we scale model size equally with the number of training tokens (Hoffmann et al., 2022). For each MatLM granularity, we also train a corresponding baseline vanilla Transformer model. That is, for each model size we train Baseline-XL, L, M, S with $d_{ff} = 4 * d_{model}, 2 * d_{model}, d_{model}, d_{model}/2$. All models have 16 layers, 16 attention heads, and a $d_{model} : d_{ff}$ ratio of $1 : 4$. We train a 256k vocabulary using the SentencePiece library (Kudo & Richardson, 2018), use a maximum context length of 1024 tokens, and a batch size of 1M tokens. We pretrained the 2.6B on 256 v3 TPU chips. We provide further details on these models in Table 3. For further details on training data, we point the reader to (Thoppilan et al., 2022).

Table 3: Model details for the models scales used to conduct the experiments described in Section 4.1, with a breakdown of total parameter counts, non-embedding parameter counts and FFN parameter counts for each model granularity.

| Parameter Count (full / spliced) | Non-Embedding Params (full / spliced) | FFN Params (full) | $d_{model}$ | N(tokens) |
|---|---|---|---|---|
| 78M (74M / 72M / 71M) | 12.6M (8.4M/6.3M/ 5.3M) | 8.4M | 256 | 10B |
| 180M (164M / 157M / 152M) | 50M (33.7M/25.3M/21.1M) | 33.6M | 512 | 20B |
| 310M (272M / 253M / 244M) | 113M (75M/56M/47M) | 75.6M | 768 | 30B |
| 463M (397M / 363M / 346M) | 201M (134M/100M/84M) | 134M | 1024 | 40B |
| 850M (696M / 620M / 582M) | 453M (302M/227M/189M) | 302M | 1536 | 80B |
| 1.3B (1B / 927M / 860M) | 805M (537M/403M/335M) | 537M | 2048 | 120B |
| 2.6B (2B / 1.7B / 1.54B) | 1.8B (1.2B/0.9B/0.7B) | 1.2B | 3072 | 160B |

## A.2    DOWNSTREAM EVALUATION

We evaluate all the LM models trained on set of 26 English tasks similar to (Brown et al., 2020; Du et al., 2022; Chowdhery et al., 2022; Anil et al., 2023), including:

1. **Open-Domain Closed-Book Question Answering tasks**: TriviaQA (Joshi et al., 2017), Natural Questions (Kwiatkowski et al., 2019), and WebQuestions (Berant et al., 2013).

2. **Cloze and completion tasks:** LAMBADA (Paperno et al., 2016), HellaSwag (Zellers et al., 2019), and StoryCloze (Mostafazadeh et al., 2016).

3. **Winograd-style tasks:** Winograd (Levesque et al., 2012) and WinoGrande (Sakaguchi et al., 2019).

4. **Reading comprehension:** SQuAD v2 (Rajpurkar et al., 2018) and RACE (Lai et al., 2017).

5. **Common sense reasoning:** PIQA (Bisk et al., 2019), ARC (Clark et al., 2018), and Open-BookQA (Mihaylov et al., 2018).

6. **SuperGLUE** (Wang et al., 2020a)

7. **Natural language inference:** Adversarial NLI (Nie et al., 2020).

Among all the downstream datasets, we classify LAMBADA, Natural Questions, SQuAD v2, WebQuestions, and TriviaQA under "GEN" tasks as these require generating a few tokens, and the remaining tasks under "RANK" tasks as they consist of choosing an option among the choices given along with the input. For all the granularities corresponding to each model, we present evaluation numbers along with development set log perplexity loss on all the 26 tasks in Tables 9 to 15. We also perform evaluation on 2.6B Mix'n'Match models and provide it in Table 16.

# B    TRAINING AND INFERENCE COSTS

We currently make minimal changes and optimizations to the training scripts of vanilla Transformer architecture. In other words, we use the same code for both Baselime and MatFormer, except using different sized splices of FFN block for each forward pass. Note that this implementation is suboptimal, as it involves added communication costs of FFN weight matrices when using model parallel

Table 4: 2.6B MatLM and Baseline training time per step, GFLOPs per step, and forward pass latencies. Each model is trained on 256 v3 TPU chips. Note that MatLM Fwd pass latency for any granularity will be same as corresponding Baseline granularity latency.

| Model | Time (s) / step | GFLOPs / step | Fwd pass latency (s) |
|---|---|---|---|
| MatLM | 2.326 | 470841 | - |
| Baseline-XL | 0.728 | 186884 | 0.234 |
| Baseline-L | 0.670 | 147317 | 0.215 |
| Baseline-M | 0.652 | 125517 | 0.198 |
| Baseline-S | 0.630 | 117556 | 0.190 |

training (discussed in more details in Appendix B.1). Though using a suboptimal implementation, we achieve the wall-clock time for MatLM training $\sim 15\%$ less to sum of wall-clock times to train all the 4 granulatities baseline counterparts. We also note that at train time, the peak memory usage is roughly equal to the sum of memory usage for the independently trained baselines. On the other hand, at inference time, both baseline and MatFormer have the same memory footprint. We give exact FLOP count, wall-clock time, and forward pass time (inference cost) of each baseline and MatLM 2.6B model (or its corresponding smaller granularities) in Table 4. During serving, we observe the 2.6B model FFN latency to attention latency ratio $= 56 : 44$. We note that this FFN:MHA latency ratio depends highly on scale and sequence length. More specifically, for a given sequence length FFN latency dominates the overall latency at scale, while the attention heads' cost increases with sequence length. We refer the reader to Kim et al. (2023) for a more extensive illustration of this. We emphasize that though we trained one MatFormer and compare its training time with Baselines combined, we get many more models than the 4 model granularities we explicitly trained.

## B.1 IMPROVING MATFORMER TRAINING EFFICIENCY

While MatFormer training uses asymptotically $2\times$ FLOPs compared to a regular Transformer, optimizations are necessary to also realize a $2\times$ runtime training performance. We discuss a few strategies here, leaving exact experimental testing to future work.

**Delayed gradient synchronization via local accumulation.** Since multiple forward and backward passes are made for each mini-batch in common implementations of data parallelism, this induces a gradient synchronization across all device for each backward pass with additional gradient accumulation. As such, for MatFormers a minimum of $2\times$ the parameters worth of gradients are exchanged for the MLP layers, thus increasing the communication overhead. Additionally, for some frameworks, such as PyTorch, gradients of the full-weight matrix size need to be exchanged, leading to $4\times$ more communication for our default experimental setup. A more efficient way to communicate gradients is to keep a local gradient accumulation buffer, which is used to accumulate all gradient from all subnetworks into the main, full-sized weight gradient. After all forward-backward passes have been completed, synchronization of gradients – with additional overall of computation and communication – can ensue. This saves $2\times$ communication overhead, reducing communication overhead to the same cost as a regular Transformer.

**Fused MatFormer kernels.** Depending on the accelerator (GPU/TPU), the smallest MatFormer forward and backward pass can be inefficient in that the matrices are too small to fully utilize the accelerator. To improve utilization at the cost of additional memory for activations, it is possible to run the following computational fusion strategy for MatFormer computation: (a) duplicate mini-batch $4\times$, (b) do the forward/backward pass for each layer for all MatFormer stages at the same time, (c) in doing so, load the tile for the weight matrix once, and reuse it for all relevant MatFormer stages. This strategy is similar to tiling strategies in FlashAttention (Dao et al., 2022) or convolution (Krizhevsky, 2009) which increase the arithmetic intensity for small weights by reusing of matrix multiplication tiles written to SRAM.

## B.2 SPECULATIVE DECODING ATTENTION SHARING

An additional benefit of MatLM is that the attention cache is shared between the draft and verifier model. When the XL model verifies S model's draft, it overwrites the attention cache with its richer

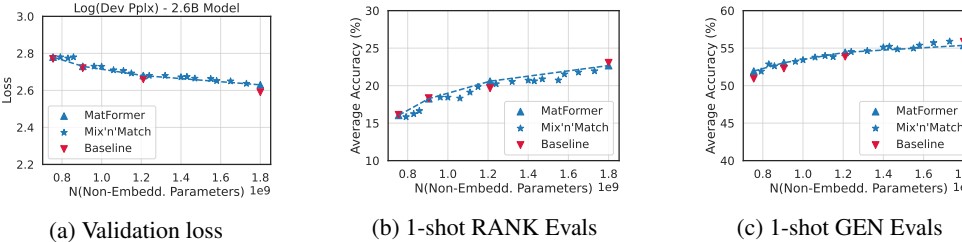

(a) Validation loss   (b) 1-shot RANK Evals   (c) 1-shot GEN Evals

Figure 6: Validation loss & one-shot downstream evaluation scores for the 2.6B MatLM & baseline models. Mix'n'Match helps generate accurate models from MatLM that lie on the performance-vs-compute curve spanned by the explicitly optimized submodels.

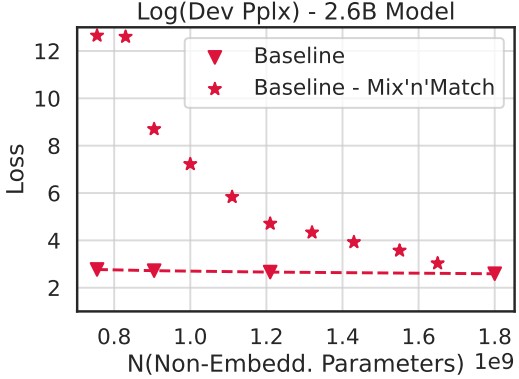

Figure 7: Validation loss for the 2.6B baseline models and their Mix'n'Match counterparts. Unlike MatLM, these extracted subnetworks perform poorly.

latent representation compared to the one generated by the drafter model. Note that 1) this does not involve extra computation since MatLM has a single universal model including both draft and verifier model; 2) attention sharing isn't possible in the Baseline since they are not explicitly trained together. Hence, latent representation of one model is quite meaningless to the other model. Thus, attention sharing gives further improvement over vanilla speculative decoding as shown in Table 1.

## C  MIX'N'MATCH

To implement Mix'n'Match, we experimented with several heuristics to select the best subnetwork, but consistently observed that gradually using larger granularities in deeper layers worked the best. More formally, we use non-decreasing hidden dimensions with the least slope (change in hidden dimensions across consecutive layers) across layers. Given that this choice behaves nearly optimally (performance lies on the pareto-optimal curve), we did not focus on search techniques. For completeness, we have plotted additional extracted subnetworks (in addition to what we have plotted in Figure 2) in Figure 6. These additional datapoints follow a similar trend. In Figure 7, we plot the validation loss of applying Mix'n'Match to vanilla Transformer baselines, and find the ability to Mix'n'Match granularities is restricted to MatLMs. In future work, we plan to extend the nested substructure to other components of the Transformer - attention heads, model dimensions, and n(layers). This would combinatorially expand the search space, warranting the use of more advanced search methods. We leave this exploration to future work.

## D  SCALING LAWS FOR LANGUAGE DECODERS

We provide results split by granularities for validation loss, average score on RANK tasks, average score on GEN tasks, and consistency in Figures 9, 10, 11, and 12 respectively. We observe that while the gap in validation loss between MatLMs and Baselines appears to be constant, the gap for downstream evaluations reduces with scale - in fact, granularities L, M and S have better downstream performance for models larger than 1B. For consistency, the gap appears to reduce with scale, but

one would need to scale the models by many orders of magnitude beyond what's possible today for baselines to have comparable consistency with MatLMs.

## D.1 SCALING LAWS OF MATFORMERS VS TRANSFORMERS.

Scaling laws are essential tools to estimate optimality under as the cost of training or inference is increased. Scaling laws can take diverse viewpoints such as overall training cost in FLOPS, training data and parameter efficiency, and inference mean FLOPS utilization vs latency for deployments.

The scaling relationship of MatFormers versus Transformers is both simple and complex. Simple, because MatFormers scaling curves for pretraining are only slightly offset from Transformers – thus MatFormers only require a fixed relative amount of additional compute and the same hyperparameters that work for Transformers are effective for MatFormers. For the setting where we use the same hyperparameters as Transformers, MatFormers need at most $10 - 20\%$ more training tokens to reach the same loss as a regular Transformer. Initial experiments where we tune hyperparameters for the individual forward/backward passes and by performing more careful initialization of the subslices the gap appears to shrink. While we do not have enough data to make definite statements, it appears MatFormer scaling can be improved to be close to Transformers scaling needing less than $0 - 5\%$ additional training tokens.

The complex scaling relationship comes from the fact that MatFormers allow the training of multiple models with a single training run which is a qualitative different from Transformers and difficult to factor into scaling equations. Essentially, in terms of efficiency, if we compare the training FLOPs equivalent of all the extractable models from MatFormers, then MatFormer training alone has a clear advantage in any case where all parameters used to train standard Transformer models on the same dataset exceed $2.58P$, where $P$ is the number of parameters of the MatFormer and the largest Transformer model. This is so because MatFormers use 2.58 times more FLOPs per token for a training run than a Transformers: $4\times$ more FLOPs for attention layers parameters and $\{1 + 1/2 + 1/4 + 1/8 = 1.875\}\times$ more FLOPs for MLP layers.

## E FURTHER ANALYSIS ON LANGUAGE DECODERS

### E.1 KL DIVERGENCE BETWEEN S, M, L AND XL MODELS

Figure 8 showcases the smoother consistency calculation between two generative models measured with KL-divergence of the smaller model's outputs with the larger model outputs. Similar to the exact match style hard consistency metric used in the main paper, there is a significant gap between the consistency of MatLM's submodels with the MatLM-XL model and between that of the corresponding baseline models. This points to how sampling strategies based on the output probabilities do not change the behavioral consistency between two models and that it still follows the trend of generating the token with the highest probability. This smoother notion of consistency argues for the metric-space preservation given that the output classifier/embedding matrix is shared across all the submodels of MatLM.

### E.2 ABLATIONS ON TRAINING METHOD

We experiment with several aspects of our training method on a 850M parameter MatLM. Our training procedure is unique compared to others (further discussed in Section 2) in 2 ways: (a) we learn all granularities in the same weight space and (b) we use joint optimization as described in Section 3. To assess the effect of these differences on performance, first we train a Transformer model with independent FFN modules with {S, M, L, XL} granularites using joint optimization (Independent modules). Next, we train a MatLM model with the only difference being that at each step, we optimize for a single granularity chosen uniformly at random (Subsampling). We find that joint optimizing a MatLM performs significantly better than these baselines, implying efficacy of both aspects of our training method.

We discuss additional ablations such as re-weighting losses to improve the performance of the XL model in Appendix E.4, and additionally studied scaling trends for these ablations. We found the reweighting loss trick to be especially powerful, bringing the performance on downstream evals

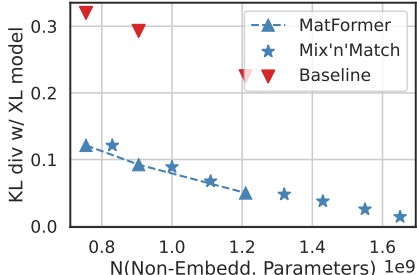

Figure 8: The smoother variant of consistency measures the KL divergence between the smaller models and the corresponding XL model. This metric, unlike the exact match accuracy variant, also accounts for different sampling strategies on the output distribution during deployment. In this figure, we plot KL divergence of S, M, L granularities with respect to XL for the 2.6B parameter model.

Table 5: We compared the validation loss of models from Joint Optimization to training MatLMs with independent MLP modules for each granularity (Independent modules) and sampling a single granularity to optimize for at each step (Subsampling) for 850M parameter models. We find that Joint Optimization performs significantly better than both these methods.

| Model | Training Strategy | XL | L | M | S |
|---|---|---|---|---|---|
| Baseline | - | 2.840 | 2.910 | 2.9710 | 3.017 |
| MatFormer | Joint Optimization | 2.874 | 2.928 | 2.980 | 3.030 |
| | Independent MLP modules | 2.894 | 2.942 | 2.985 | 3.030 |
| | Subsampling | 2.929 | 2.946 | 2.999 | 3.049 |

within $0.1\%$ for the XL model. This also nudges us towards finding better hyperparameters and weight initializations for reliable scaling of MatLMs (Yang et al., 2022).

### E.3 CHANGING EMBEDDING SIZE

Because of the ubiquity of 64k vocabs size (Brown et al., 2020) we additionally train models upto 201M non-embedding parameters similar to those described in Appendix A, except that the embedding size is 64k (the largest model corresponds to the 463M parameter model). We plot the scaling trends in Figure 13. Though 4 models is not enough to extrapolate a trend, we observe that the scaling trend for validation loss appears to be similar.

### E.4 REWEIGHTING STRATEGIES

We additionally experiment with reweighting the losses for the individual granularities in order to boost the performance of the largest granularity while minimally impacting the performance of the smaller granularities. We present the relative weights used in Table 6 as $\lambda_4 : \lambda_3 : \lambda_2 : \lambda_1$, and find that in general, upweighting the largest granularity greatly improves quality. Another interesting related direction for improving MatFormer performance further is granularity appropriate initialization (Yang et al., 2022).

### E.5 SCALING LAWS FOR REWEIGHTED STRATEGY

We conduct scaling experiments similar to those described in Section 4.1 for the reweighed models, specifically for models with the ratio $2 : 1.5 : 1.25 : 1$, and plot the results in Figure 14. We note that the scaling trend is similar to the MatLM with a $1 : 1 : 1 : 1$ relative weighting ($a = 19.889, b = -0.130, c = 1.374$), but with a slightly better validation loss .

Table 6: For 850M model, we experiment with modifying $\mathcal{L}_{\text{JOINT}}$ to use a weighted average as opposed to an unweighted average, and report the results across all granularities. We find that all strategies that upweight the loss for the largest granularity perform well, with modest degradation on the M and S granularties.

| Model | Relative Weights | XL | L | M | S |
|---|---|---|---|---|---|
| Baseline | N/A | 2.840 | 2.910 | 2.971 | 3.017 |
| MatFormer | 1:1:1:1 | 2.874 | 2.928 | 2.980 | 3.030 |
| | $2 : 1.5 : 1.25 : 1$ | 2.867 | 2.927 | 2.986 | 3.051 |
| | $1 : 1.25 : 1.5 : 2$ | 2.883 | 2.936 | 2.982 | 3.026 |
| | $2 : 1 : 1 : 1$ | 2.863 | 2.929 | 2.985 | 3.043 |
| | $\sqrt{8} : \sqrt{4} : \sqrt{2} : 1$ | 2.862 | 2.924 | 2.990 | 3.063 |

# F FURTHER ANALYSIS ON VISION ENCODERS

## F.1 DECOUPLING EFFECT OF MATFORMER ON PRETRAINING AND FINETUNING

Table 7 investigates the effect of MatFormer on pretaining and finetuning phases of ViT-L/16 model. ViT-L/16 is typically pretrained on ImageNet-21K and then finetuned on ImageNet-1K for the final evaluation. Table 7 shows that having a MatFormer during pretraining generates a better model for downstream finetuning compared to regular ViT pertaining. At the same time, finetuning a vanilla pretrained ViT with MatFormer results in flexibility being induced into the model. Despite being up to 2% less accurate than its counterparts at some granularities, a fine-tuned MatViT learned to reallocate the information to provide strong nested models. Considering that this is insignificant compared to pretaining costs, possible to take the largest pretrained ViT model and finetune with MatFormer to obtain a deployable MatViT variant.

Table 7: $2 \times 2$ grid of pairs to evaluate (top-1 accuracy (%)) the effects of MatFormer and standard training on the pretraining (PT) on ImageNet-21K and finetuning (FT) on ImageNet-1K using a L/16 architecture. Using a MatFormer during pretraining helps bring more accurate, and elastic encoders for downstream uses.

| PT↓ / FT→ | # Params (M) | ViT | MatViT |
|---|---|---|---|
| ViT | 306 | 85.26 | 85.57 |
| | 206 | 85.12 | 84.27 |
| | 156 | 85.02 | 82.79 |
| | 131 | 84.42 | 82.1 |
| MatViT | 306 | 85.58 | 85.61 |
| | 206 | – | 85.40 |
| | 156 | – | 85.02 |
| | 131 | – | 84.41 |

## F.2 TRADITIONAL IMAGE RETRIEVAL EVALUATION

Table 8 showcases traditional image retrieval evaluation on ImageNet-1K where the query and the document encoders are the same for nearest neighbor retrieval. The 1-nearest neighbor (NN) based evaluation closely follows one-vs-all classification results shown in Figure 4. Both MatViT variants B/16 and L/16 have submodels that have as good or better retrieval performance compared to their independently trained counterparts. Concretely, MatViT-based retrieval can be up to 0.5% more accurate than the baselines while a 200M parameter MatViT submodel can be more accurate than the 300M parameter ViT baseline.

Table 8: Image retrieval 1-NN accuracy (%) when the query and document encoders are the same model. Similar to the image classification results, MatViT variants either match or outperform the corresponding standard ViT counterparts. Note that all the smaller models of a given model in MatViT are extracted for free while the baselines have to be explicitly trained for the constraints.

| Encoder | # Params (M) | ViT | MatViT |
|---------|--------------|-------|--------|
| B/16    | 85           | 77.46 | 77.38  |
|         | 57           | 76.58 | 76.41  |
|         | 43           | 74.90 | 74.49  |
|         | 36           | 71.44 | 71.72  |
| L/16    | 300          | 83.17 | 83.67  |
|         | 200          | 82.92 | 83.23  |
|         | 150          | 82.81 | 82.89  |
|         | 125          | 82.22 | 82.14  |

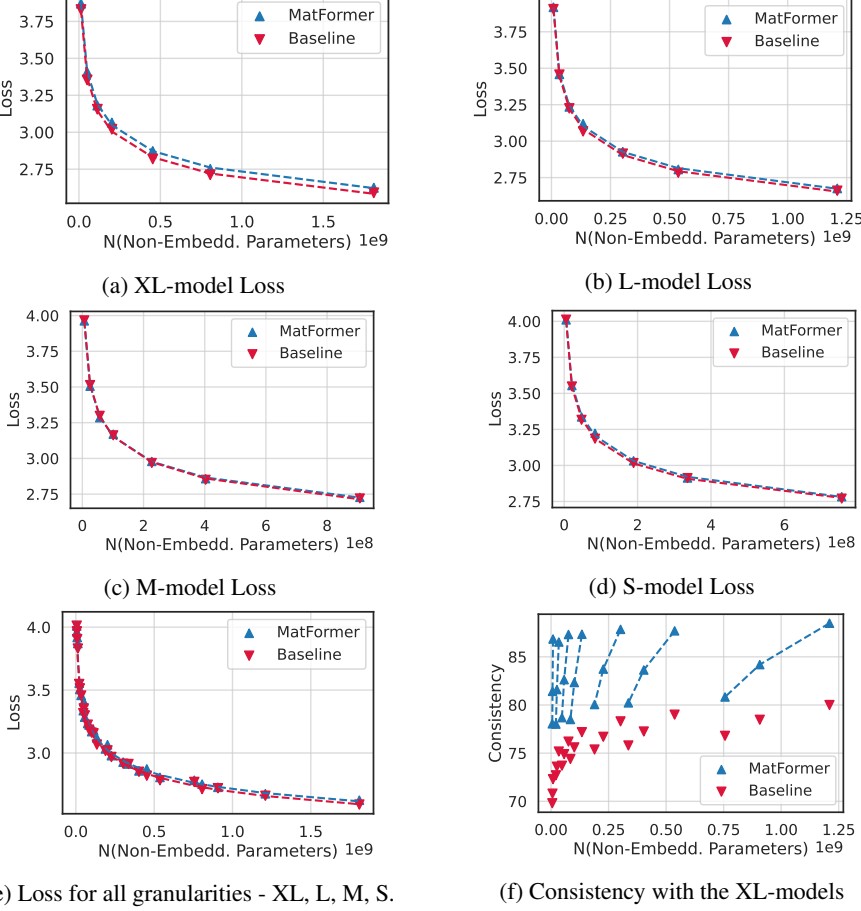

(a) XL-model Loss

(b) L-model Loss

(c) M-model Loss

(d) S-model Loss

(e) Loss for all granularities - XL, L, M, S.

(f) Consistency with the XL-models

Figure 9: We train various decoder-only MatLM models at a range of sizes from 78M to 2.6B parameters and observe the scaling trends for each model granularity on validation loss. We observe that the gap between MatLM and the baseline appears to be constant at each granularity. The consistency between the submodels of granularities and the XL models shows the effect of MatFormer joint training on natively ensuring similar behavior across submodels.

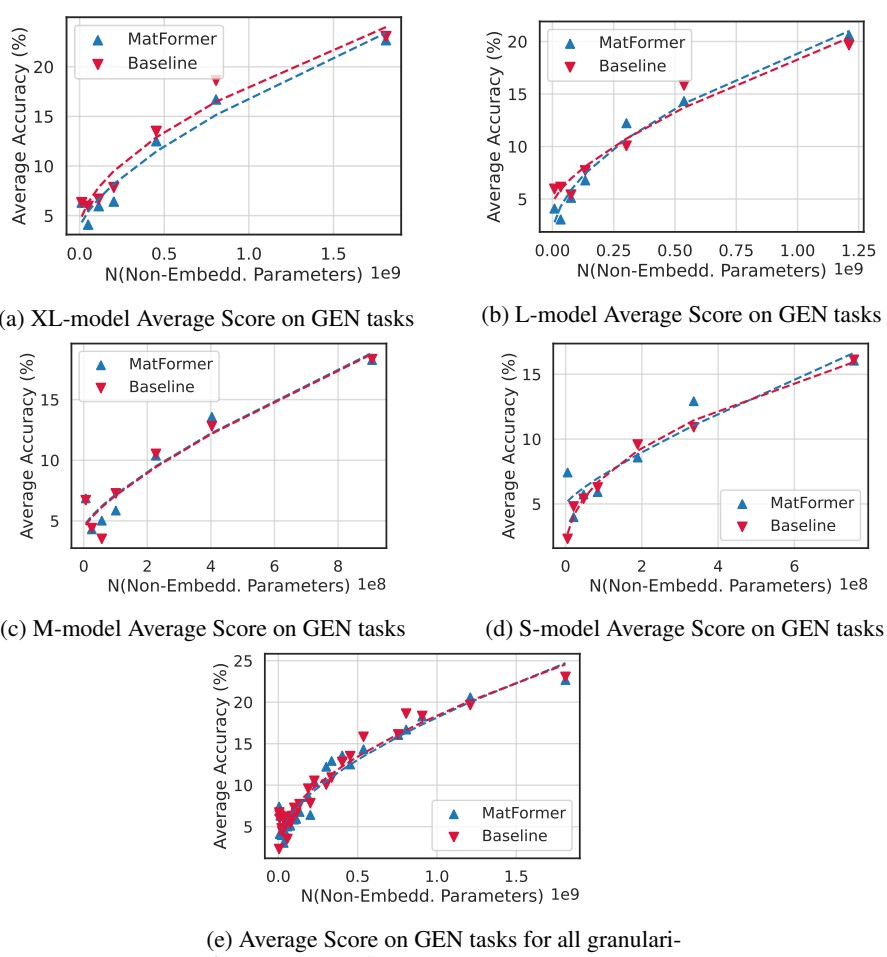

(a) XL-model Average Score on GEN tasks

(b) L-model Average Score on GEN tasks

(c) M-model Average Score on GEN tasks

(d) S-model Average Score on GEN tasks

(e) Average Score on GEN tasks for all granularities - XL, L, M, S.

Figure 10: We train various decoder-only MatLM models at a range of sizes from 78M to 2.6B parameters and observe the scaling trends for each model granularity for the average score on GEN tasks 1-shot evaluation. We observe that the gap between MatLM and the baseline reduces with scale, outperforming the baselines for S, M, L granularities for the largest models.

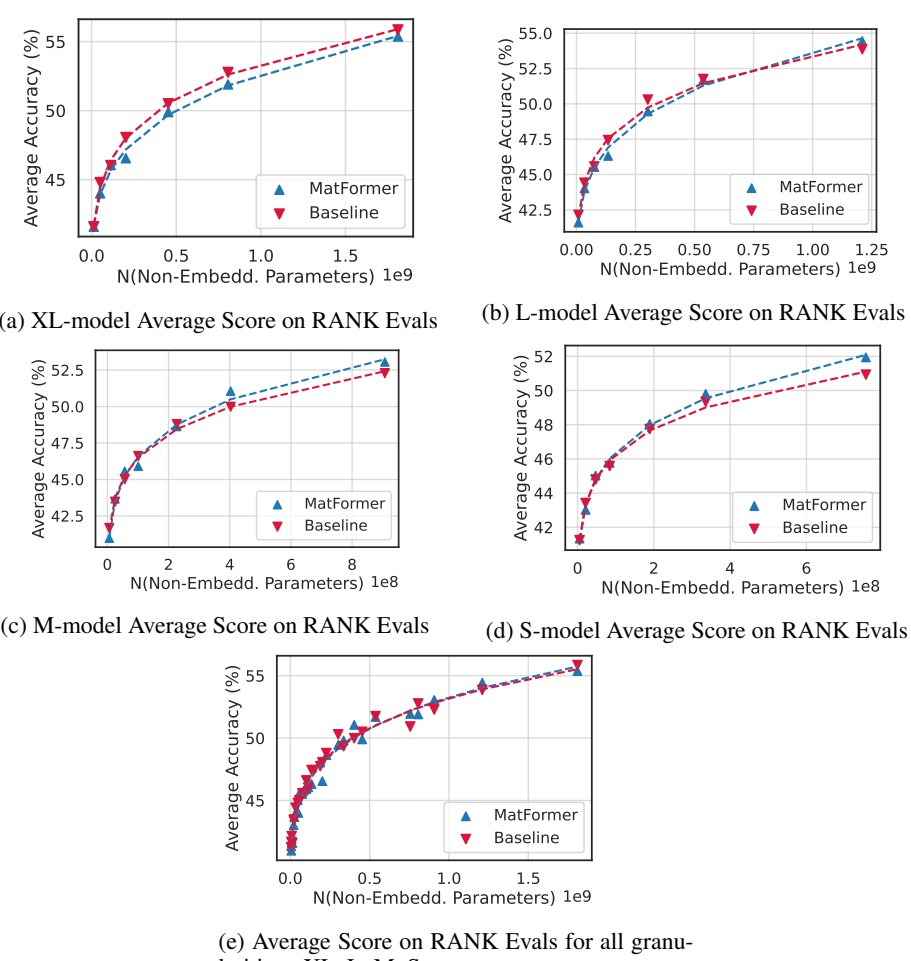

(a) XL-model Average Score on RANK Evals

(b) L-model Average Score on RANK Evals

(c) M-model Average Score on RANK Evals

(d) S-model Average Score on RANK Evals

(e) Average Score on RANK Evals for all granu-larities - XL, L, M, S

Figure 11: We train various decoder-only MatLM models at a range of sizes from 78M to 2.6B parameters and observe the scaling trends for each model granularity for the average score on RANK 1-shot evaluation. We observe that the gap between MatLM and the baseline reduces with scale, outperforming the baselines for S, M, L granularities for the largest models.

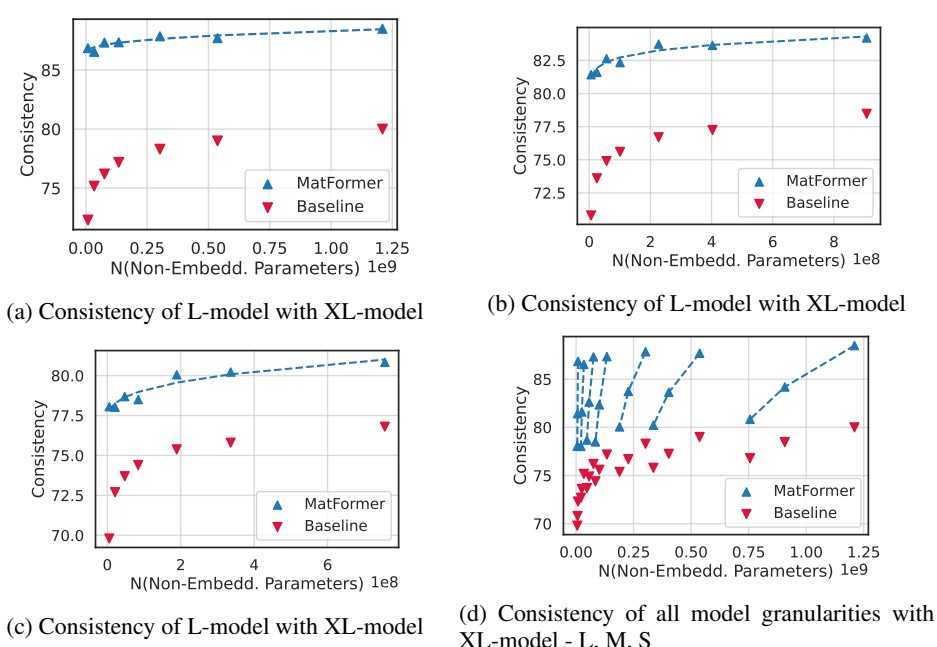

(a) Consistency of L-model with XL-model

(b) Consistency of L-model with XL-model

(c) Consistency of L-model with XL-model

(d) Consistency of all model granularities with XL-model - L, M, S

Figure 12: We train various decoder-only MatLM models at a range of sizes from 78M to 2.6B parameters and observe the scaling trends for each submodel S, M, L for the consistency with the XL model. We observe that the gap between MatLM and the baseline reduces with scale, but one would need to scale the baseline by many orders of magnitude to have consistency comparable to that of MatLMs.

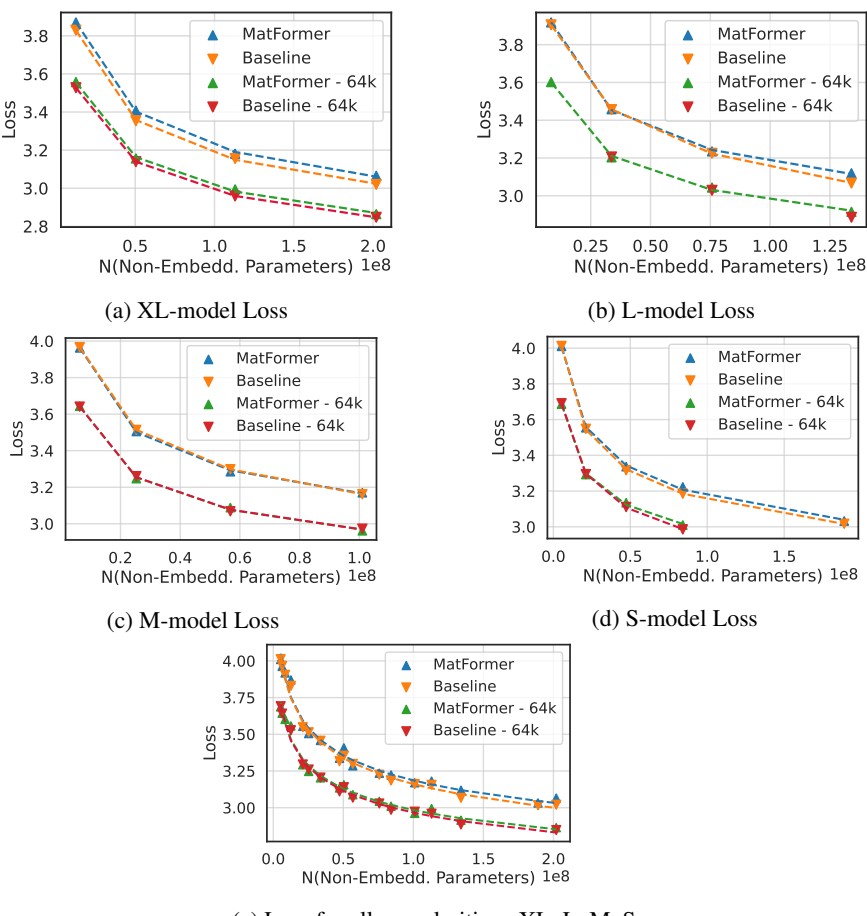

(a) XL-model Loss

(b) L-model Loss

(c) M-model Loss

(d) S-model Loss

(e) Loss for all granularities - XL, L, M, S

Figure 13: We train various decoder-only MatLM models at a range of sizes from 29M to 267M parameters with an embedding size of 64k and observe the scaling trends for each model granularity on validation loss. We observe that the gap between MatLM and the baseline appears to be constant at each granularity, similar to what is observed in Figure 9.

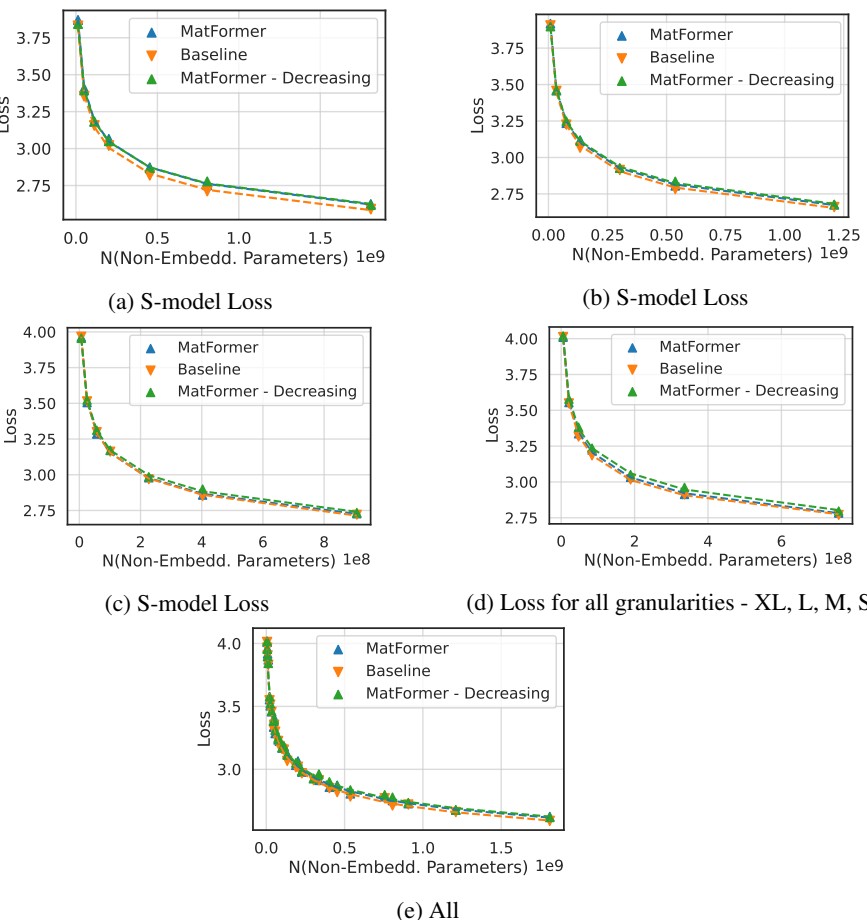

(a) S-model Loss

(b) S-model Loss

(c) S-model Loss

(d) Loss for all granularities - XL, L, M, S

(e) All

Figure 14: We train various decoder-only MatLM models at a range of sizes from 78M to 2.6B parameters with a reweighing ratio of $2 : 1.5 : 1.25 : 1$ and observe the scaling trends for each model granularity on validation loss. We observe that the gap between MatLM and the baseline appears to be constant at each granularity, similar to what is observed in Figure 9.

Table 9: Downstream Eval numbers and development set log perplexity loss on 78M model size granularities.

| Downstream Task | Baseline-S | MatLM-S | Baseline-M | MatLM-M | Baseline-L | MatLM-L | Baseline-XL | MatLM-XL |
|---|---|---|---|---|---|---|---|---|
| TriviaQA (EM) | 0.14 | 0.16 | 0.19 | 0.25 | 0.14 | 0.3 | 0.19 | 0.28 |
| NaturalQuestions (EM) | 0.06 | 0.03 | 0.03 | 0.06 | 0.03 | 0.03 | 0.03 | 0.03 |
| WebQuestions (EM) | 0.1 | 0.2 | 0.15 | 0.2 | 0.2 | 0.3 | 0.3 | 0.3 |
| LAMBADA | 0.06 | 0.02 | 0.02 | 0 | 0.02 | 0 | 0 | 0 |
| HellaSwag | 25.42 | 26.28 | 26 | 25.87 | 25.95 | 25.9 | 25.95 | 25.94 |
| StoryCloze | 52.81 | 53.39 | 53.13 | 53.34 | 54.46 | 53.5 | 54.46 | 54.36 |
| WSC | 52.98 | 51.93 | 53.68 | 50.88 | 55.79 | 54.04 | 52.28 | 52.63 |
| WinoGrande | 48.46 | 51.54 | 51.54 | 47.99 | 50.99 | 48.46 | 48.86 | 49.41 |
| Winograd | 53.11 | 52.75 | 52.38 | 53.85 | 55.31 | 55.31 | 52.75 | 55.68 |
| SQuAD v2 (EM) | 11.19 | 36.71 | 33.14 | 33.77 | 20.08 | 29.17 | 22.78 | 30.97 |
| RACE-H | 25.53 | 25.84 | 24.73 | 25.44 | 26.07 | 25.9 | 25.96 | 25.84 |
| RACE-M | 29.18 | 30.15 | 28.83 | 29.94 | 28.83 | 30.43 | 29.74 | 31.48 |
| PIQA | 55.77 | 55.22 | 54.62 | 55.28 | 54.52 | 54.79 | 56.86 | 54.08 |
| ARC-C | 21.5 | 20.9 | 21.08 | 21.67 | 21.59 | 21.33 | 22.35 | 22.1 |
| ARC-E | 34.55 | 35.48 | 34.3 | 35.73 | 34.89 | 36.11 | 34.55 | 35.98 |
| OpenBookQA | 25.4 | 28.6 | 27.6 | 28 | 28.2 | 28 | 29.8 | 29 |
| BoolQ | 48.72 | 44.89 | 51.87 | 47.37 | 51.28 | 46.85 | 52.11 | 45.87 |
| COPA | 62 | 64 | 62 | 61 | 63 | 63 | 60 | 60 |
| RTE | 53.79 | 52.35 | 52.35 | 51.99 | 51.26 | 54.51 | 51.99 | 52.71 |
| WiC | 49.53 | 47.34 | 49.06 | 47.34 | 47.34 | 47.34 | 47.65 | 47.34 |
| MultiRC (F1) | 53.17 | 51.72 | 53.42 | 53.28 | 56.86 | 53.82 | 55.46 | 53.42 |
| ReCoRD | 39.52 | 39.22 | 40.03 | 39.95 | 40.55 | 40.42 | 40.8 | 40.83 |
| CB | 41.07 | 42.86 | 44.64 | 39.29 | 44.64 | 41.07 | 42.86 | 44.64 |
| ANLI-R1 | 30.9 | 32 | 32.3 | 31.9 | 32.5 | 32.3 | 32.5 | 31.7 |
| ANLI-R2 | 31.1 | 30.9 | 31.1 | 30.1 | 30.7 | 30.8 | 30.6 | 30.3 |
| ANLI-R3 | 31.75 | 30.75 | 30.58 | 30.25 | 30.33 | 29.67 | 30 | 30.17 |
| Average | 33.76 | 34.82 | 34.95 | 34.41 | 34.83 | 34.74 | 34.65 | 34.81 |
| Avg over GEN Taks | 2.31 | 7.42 | 6.7 | 6.85 | 4.09 | 5.96 | 4.66 | 6.31 |
| Avg over RANK Tasks | 41.25 | 41.34 | 41.68 | 40.97 | 42.15 | 41.6 | 41.79 | 41.59 |
| Dev set log pplx | 4.010 | 4.012 | 3.97 | 3.96 | 3.905 | 3.908 | 3.83 | 3.868 |

Table 10: Downstream Eval numbers and development set log perplexity loss on 180M model size granularities.

| Downstream Task | Baseline-S | MatLM-S | Baseline-M | MatLM-M | Baseline-L | MatLM-L | Baseline-XL | MatLM-XL |
|---|---|---|---|---|---|---|---|---|
| TriviaQA (EM) | 1.04 | 0.9 | 0.98 | 1.26 | 1.16 | 1.89 | 1.86 | 2.00 |
| NaturalQuestions (EM) | 0.08 | 0.11 | 0.14 | 0.08 | 0.3 | 0.11 | 0.28 | 0.11 |
| WebQuestions (EM) | 0.59 | 0.94 | 0.44 | 0.98 | 1.28 | 0.89 | 1.33 | 0.79 |
| LAMBADA | 0.16 | 0.68 | 0.43 | 1.16 | 1.51 | 0.95 | 0.49 | 0.99 |
| HellaSwag | 27.77 | 27.3 | 27.45 | 27.61 | 27.58 | 27.84 | 28.86 | 28.56 |
| StoryCloze | 56.33 | 56.07 | 57.03 | 56.87 | 57.3 | 57.78 | 58.63 | 58.52 |
| WSC | 55.44 | 55.44 | 56.49 | 60.35 | 58.25 | 58.6 | 57.54 | 58.6 |
| WinoGrande | 52.01 | 50.12 | 50.28 | 49.17 | 51.22 | 50.43 | 51.54 | 49.09 |
| Winograd | 54.21 | 55.68 | 56.78 | 57.51 | 61.54 | 58.61 | 60.44 | 61.17 |
| SQuAD v2 (EM) | 22.13 | 17.28 | 20.05 | 18.02 | 26.42 | 11.42 | 25.76 | 16.53 |
| RACE-H | 27.93 | 27.9 | 27.5 | 28.53 | 28.7 | 28.82 | 28.73 | 28.73 |
| RACE-M | 33.29 | 34.47 | 34.19 | 34.05 | 34.54 | 33.91 | 33.29 | 34.19 |
| PIQA | 57.13 | 58.05 | 56.91 | 57.94 | 57.94 | 58.00 | 59.52 | 58.92 |
| ARC-C | 22.53 | 22.61 | 23.63 | 22.27 | 24.06 | 22.1 | 24.66 | 23.55 |
| ARC-E | 40.24 | 39.39 | 40.19 | 40.49 | 41.71 | 40.74 | 41.62 | 41.16 |
| OpenBookQA | 30.60 | 31.00 | 30.80 | 31.80 | 31.00 | 32.80 | 34.00 | 32.6 |
| BoolQ | 54.13 | 52.23 | 52.45 | 52.05 | 55.63 | 52.17 | 55.9 | 48.44 |
| COPA | 62 | 61 | 61 | 61 | 61 | 64 | 64 | 65 |
| RTE | 52.71 | 53.07 | 52.35 | 53.43 | 50.54 | 52.71 | 52.71 | 52.71 |
| WiC | 47.34 | 51.41 | 47.34 | 49.37 | 47.96 | 47.81 | 47.65 | 47.34 |
| MultiRC (F1) | 54.34 | 53.34 | 45.65 | 56.12 | 47.47 | 52.62 | 47.62 | |
| ReCoRD | 48.58 | 49.4 | 48.99 | 50.13 | 50.56 | 51.25 | 52.82 | 52.51 |
| CB | 42.86 | 44.64 | 42.86 | 44.64 | 39.29 | 44.64 | 42.86 | 42.86 |
| ANLI-R1 | 31.8 | 32.6 | 31.8 | 32.4 | 32.4 | 32.8 | 32.2 | 32.1 |
| ANLI-R2 | 30.5 | 29.8 | 31.1 | 29.8 | 32.00 | 30.5 | 30.5 | 30.1 |
| ANLI-R3 | 30.08 | 30.25 | 30.5 | 32.00 | 33.5 | 31.42 | 30.67 | 30.42 |
| Average | 35.99 | 35.51 | 35.96 | 36.1 | 37.06 | 36.14 | 37.33 | 36.33 |
| GPT3-GEN | 4.8 | 3.98 | 4.41 | 4.3 | 6.14 | 3.05 | 5.94 | 4.08 |
| GPT3-RANK | 43.42 | 43.02 | 43.48 | 43.67 | 44.42 | 44.02 | 44.8 | 44.01 |
| Dev set log pplx | 3.55 | 3.55 | 3.512 | 3.505 | 3.456 | 3.458 | 3.354 | 3.40 |

Table 11: Downstream Eval numbers and development set log perplexity loss on 310M model size granularities.

| Downstream Task | Baseline-S | MatLM-S | Baseline-M | MatLM-M | Baseline-L | MatLM-L | Baseline-XL | MatLM-XL |
|---|---|---|---|---|---|---|---|---|
| TriviaQA (EM) | 2.09 | 2.4 | 2.2 | 3.17 | 2.84 | 2.73 | 5.18 | 3.12 |
| NaturalQuestions (EM) | 0.11 | 0.28 | 0.28 | 0.5 | 0.58 | 0.3 | 0.91 | 0.61 |
| WebQuestions (EM) | 2.12 | 1.38 | 1.08 | 1.67 | 1.67 | 1.43 | 2.41 | 1.57 |
| LAMBADA | 0.29 | 1.79 | 0.66 | 1.92 | 1.9 | 2.46 | 2.76 | 2.64 |
| HellaSwag | 29.89 | 29.69 | 30.05 | 30.02 | 31.18 | 30.63 | 32.52 | 31.58 |
| StoryCloze | 59.17 | 58.85 | 59.54 | 60.13 | 60.24 | 60.5 | 61.68 | 61.36 |
| WSC | 61.05 | 59.65 | 59.3 | 58.6 | 61.75 | 56.84 | 58.95 | 57.19 |
| WinoGrande | 51.46 | 52.88 | 49.57 | 50.91 | 52.41 | 50.75 | 50.91 | 52.01 |
| Winograd | 55.68 | 56.04 | 57.88 | 59.71 | 63 | 59.71 | 61.17 | 60.07 |
| SQuAD v2 (EM) | 22.38 | 22.79 | 13.38 | 17.83 | 20.03 | 18.66 | 22.03 | 21.81 |
| RACE-H | 29.45 | 28.33 | 28.9 | 28.67 | 29.22 | 29.07 | 29.67 | 28.79 |
| RACE-M | 35.31 | 36.14 | 36.14 | 36.91 | 36.42 | 36.14 | 37.6 | 36.07 |
| PIQA | 58.98 | 59.9 | 59.58 | 59.85 | 59.79 | 60.45 | 62.19 | 60.61 |
| ARC-C | 23.38 | 20.82 | 23.21 | 21.33 | 23.81 | 23.21 | 25 | 22.95 |
| ARC-E | 42.3 | 42.34 | 44.11 | 43.52 | 44.53 | 44.44 | 46.8 | 45.62 |
| OpenBookQA | 32.8 | 35.2 | 34.6 | 36.4 | 35.2 | 35.8 | 36.8 | 36.6 |
| BoolQ | 53.43 | 59.05 | 55.32 | 58.72 | 52.87 | 57.22 | 54.22 | 55.6 |
| COPA | 61 | 61 | 61 | 66 | 64 | 63 | 60 | 66 |
| RTE | 52.71 | 54.51 | 53.43 | 51.62 | 51.62 | 53.07 | 54.15 | 49.46 |
| WiC | 47.18 | 48.43 | 47.65 | 49.22 | 47.65 | 50.16 | 47.34 | 51.25 |
| MultiRC (F1) | 53.07 | 51.69 | 53.5 | 51.36 | 48.46 | 47.14 | 45.72 | 46.23 |
| ReCoRD | 54.34 | 53.86 | 55.18 | 55.33 | 56.75 | 56.79 | 58.39 | 58.07 |
| CB | 42.86 | 46.43 | 42.86 | 46.43 | 42.86 | 46.43 | 50 | 51.79 |
| ANLI-R1 | 32 | 31.3 | 32 | 32.2 | 32.5 | 32.3 | 32.2 | 32.8 |
| ANLI-R2 | 32.6 | 30.2 | 30.9 | 29.8 | 30.6 | 31.2 | 29.8 | 30.9 |
| ANLI-R3 | 32.08 | 29.25 | 30.75 | 30.08 | 32.17 | 31.25 | 31.5 | 32.17 |
| Average | 37.22 | 37.47 | 37.04 | 37.77 | 37.85 | 37.76 | 38.46 | 38.34 |
| Avg over GEN Taks | 5.4 | 5.73 | 3.52 | 5.02 | 5.41 | 5.12 | 6.66 | 5.95 |
| Avg over RANK Tasks | 44.8 | 45.03 | 45.02 | 45.56 | 45.57 | 45.53 | 46.03 | 46.05 |
| Dev set log pplx | 3.31 | 3.33 | 3.30 | 3.285 | 3.224 | 3.235 | 3.15 | 3.18 |

Table 12: Downstream Eval numbers and development set log perplexity loss on 463M model size granularities.

| Downstream Task | Baseline-S | MatLM-S | Baseline-M | MatLM-M | Baseline-L | MatLM-L | Baseline-XL | MatLM-XL |
|---|---|---|---|---|---|---|---|---|
| TriviaQA (EM) | 4.63 | 3.87 | 4.87 | 4.55 | 6.11 | 5.63 | 8.09 | 6.48 |
| NaturalQuestions (EM) | 0.61 | 0.58 | 0.8 | 0.89 | 0.94 | 1.16 | 1.66 | 1.25 |
| WebQuestions (EM) | 2.31 | 1.62 | 2.26 | 2.02 | 2.85 | 2.31 | 2.85 | 2.56 |
| LAMBADA | 2.1 | 1.65 | 2.6 | 2.1 | 3.94 | 2.93 | 3.49 | 3.49 |
| HellaSwag | 32.12 | 31.57 | 32.83 | 32.16 | 33.8 | 33.48 | 36.21 | 35.08 |
| StoryCloze | 61.25 | 60.98 | 61.36 | 61.46 | 63.66 | 62.21 | 64.24 | 64.08 |
| WSC | 57.54 | 64.91 | 61.4 | 62.11 | 66.32 | 62.11 | 61.05 | 63.16 |
| WinoGrande | 52.33 | 51.38 | 49.09 | 50.99 | 52.64 | 50.36 | 53.12 | 52.64 |
| Winograd | 60.07 | 63.74 | 60.07 | 62.27 | 67.4 | 61.54 | 68.5 | 63.74 |
| SQuAD v2 (EM) | 21.7 | 21.85 | 25.8 | 19.71 | 24.69 | 21.85 | 23.08 | 18.28 |
| RACE-H | 29.85 | 29.45 | 29.47 | 29.79 | 30.56 | 29.79 | 30.7 | 30.02 |
| RACE-M | 37.53 | 37.6 | 37.33 | 38.93 | 40.39 | 39.62 | 40.95 | 39.21 |
| PIQA | 61.26 | 61.53 | 61.48 | 62.08 | 60.99 | 63.22 | 63.17 | 63.71 |
| ARC-C | 23.04 | 22.7 | 24.06 | 22.35 | 24.49 | 22.18 | 23.72 | 23.63 |
| ARC-E | 45.83 | 44.44 | 46.3 | 45.62 | 47.73 | 47.85 | 51.73 | 49.12 |
| OpenBookQA | 37.2 | 36.4 | 37 | 37.8 | 36.4 | 39.2 | 41 | 38.4 |
| BoolQ | 52.39 | 52.69 | 56.12 | 52.05 | 50.28 | 51.28 | 54.98 | 47.95 |
| COPA | 67 | 62 | 73 | 63 | 71 | 63 | 67 | 66 |
| RTE | 52.35 | 53.07 | 53.43 | 52.71 | 52.35 | 52.71 | 52.35 | 51.99 |
| WiC | 47.34 | 47.34 | 47.34 | 47.34 | 47.34 | 47.34 | 47.34 | 47.34 |
| MultiRC (F1) | 45.63 | 46.02 | 54.4 | 46.38 | 52.79 | 49.28 | 52.34 | 41.71 |
| ReCoRD | 57.58 | 58.65 | 59.31 | 59.71 | 60.87 | 61 | 63.42 | 61.77 |
| CB | 42.86 | 42.86 | 44.64 | 42.86 | 44.64 | 42.86 | 42.86 | 42.86 |
| ANLI-R1 | 32.6 | 32.5 | 31.7 | 33.1 | 31.4 | 32.3 | 32.5 | 32.6 |
| ANLI-R2 | 30.7 | 30.7 | 28.4 | 30.5 | 30.4 | 30.6 | 31.2 | 31.8 |
| ANLI-R3 | 30.83 | 30.67 | 30.08 | 30.75 | 30.83 | 30.67 | 30.92 | 30.75 |
| Average | 38.02 | 38.11 | 39.04 | 38.2 | 39.8 | 38.71 | 40.33 | 38.83 |
| Avg over GEN Taks | 6.27 | 5.91 | 7.27 | 5.85 | 7.71 | 6.78 | 7.84 | 6.41 |
| Avg over RANK Tasks | 45.59 | 45.77 | 46.61 | 45.9 | 47.44 | 46.31 | 48.06 | 46.55 |
| Dev set log pplx | 3.205 | 3.217 | 3.16 | 3.16 | 3.096 | 3.11 | 3.023 | 3.06 |

Table 13: Downstream Eval numbers and development set log perplexity loss on 850M model size granularities.

| Downstream Task | Baseline-S | MatLM-S | Baseline-M | MatLM-M | Baseline-L | MatLM-L | Baseline-XL | MatLM-XL |
|---|---|---|---|---|---|---|---|---|
| TriviaQA (EM) | 9.26 | 6.62 | 10.82 | 9.78 | 11.07 | 11.72 | 13.31 | 13.76 |
| NaturalQuestions (EM) | 1.66 | 0.89 | 1.69 | 1.58 | 2.24 | 2.38 | 2.66 | 2.74 |
| WebQuestions (EM) | 3.89 | 3.35 | 4.08 | 4.18 | 3.74 | 4.43 | 4.08 | 5.31 |
| LAMBADA | 3.2 | 8.25 | 6.97 | 10.83 | 8.19 | 10.44 | 14.03 | 10.83 |
| HellaSwag | 36.11 | 36.64 | 38.26 | 37.7 | 40.63 | 39.64 | 43.4 | 42.55 |
| StoryCloze | 64.78 | 65.26 | 66.33 | 66.17 | 68.25 | 67.13 | 71.25 | 69.64 |
| WSC | 66.32 | 65.96 | 63.16 | 64.21 | 69.82 | 69.12 | 70.53 | 68.42 |
| WinoGrande | 52.17 | 51.54 | 52.25 | 52.57 | 55.17 | 52.96 | 54.14 | 54.62 |
| Winograd | 68.13 | 69.23 | 67.03 | 71.43 | 71.06 | 70.33 | 72.16 | 72.89 |
| SQuAD v2 (EM) | 29.9 | 23.79 | 29.07 | 25.51 | 25.07 | 26.39 | 33.41 | 28.46 |
| RACE-H | 30.39 | 30.76 | 31.93 | 31.88 | 32.53 | 31.88 | 33.79 | 32.73 |
| RACE-M | 40.95 | 40.95 | 42.06 | 41.16 | 42.27 | 42.55 | 44.64 | 42.48 |
| PIQA | 64.04 | 63.98 | 64.64 | 64.91 | 65.45 | 65.23 | 67.25 | 66.21 |
| ARC-C | 24.49 | 24.15 | 26.71 | 24.91 | 26.71 | 26.54 | 27.13 | 27.47 |
| ARC-E | 52.15 | 51.01 | 53.66 | 52.95 | 56.27 | 54.92 | 57.11 | 56.57 |
| OpenBookQA | 38.2 | 40.4 | 40.8 | 41.2 | 42.8 | 40.8 | 43 | 42 |
| BoolQ | 52.63 | 50.31 | 51.9 | 47.8 | 56.73 | 50.15 | 55.6 | 48.41 |
| COPA | 68 | 73 | 68 | 73 | 71 | 73 | 73 | 76 |
| RTE | 51.62 | 51.99 | 52.71 | 52.35 | 51.62 | 51.99 | 53.07 | 52.71 |
| WiC | 47.34 | 47.18 | 47.34 | 47.18 | 47.34 | 47.18 | 47.34 | 47.18 |
| MultiRC (F1) | 44.37 | 51.32 | 52.11 | 50.46 | 54.7 | 53 | 37.58 | 47.16 |
| ReCoRD | 63.52 | 64.27 | 65.03 | 65.36 | 67.55 | 66.53 | 69.56 | 68.03 |
| CB | 42.86 | 37.5 | 42.86 | 42.86 | 42.86 | 42.86 | 46.43 | 39.29 |
| ANLI-R1 | 30.9 | 31.8 | 33.7 | 32.1 | 31.7 | 32.2 | 32.6 | 32.4 |
| ANLI-R2 | 31.8 | 31.5 | 31.5 | 30.9 | 31.1 | 30.6 | 30.4 | 30.8 |
| ANLI-R3 | 32 | 30.25 | 32.83 | 30.17 | 30.75 | 30 | 30.58 | 30.25 |
| Average | 40.41 | 40.46 | 41.44 | 41.27 | 42.56 | 42.08 | 43.39 | 42.65 |
| Avg over GEN Taks | 9.58 | 8.58 | 10.53 | 10.38 | 10.06 | 11.07 | 13.5 | 12.22 |
| Avg over RANK Tasks | 47.75 | 48.05 | 48.8 | 48.63 | 50.3 | 49.46 | 50.5 | 49.9 |
| Dev set log pplx | 3.017 | 3.03 | 2.971 | 2.98 | 2.91 | 2.928 | 2.84 | 2.874 |

Table 14: Downstream Eval numbers and development set log perplexity loss on 1.3B model size granularities.

| Downstream Task | Baseline-S | MatLM-S | Baseline-M | MatLM-M | Baseline-L | MatLM-L | Baseline-XL | MatLM-XL |
|---|---|---|---|---|---|---|---|---|
| TriviaQA (EM) | 11.92 | 12 | 14.68 | 13.09 | 16.48 | 14.91 | 20.14 | 17.62 |
| NaturalQuestions (EM) | 1.88 | 2.19 | 2.24 | 2.47 | 3.07 | 2.99 | 4.79 | 4.13 |
| WebQuestions (EM) | 3.84 | 5.02 | 4.72 | 5.36 | 5.07 | 5.76 | 6.05 | 6.15 |
| LAMBADA | 7.3 | 9.94 | 13.55 | 12.34 | 17.97 | 13.51 | 22.65 | 19.21 |
| HellaSwag | 40.53 | 40.35 | 42.86 | 42.5 | 46 | 44.48 | 49.78 | 47.69 |
| StoryCloze | 67.29 | 68.2 | 69.75 | 69.91 | 72.37 | 71.14 | 73.81 | 72.8 |
| WSC | 64.56 | 65.96 | 64.91 | 69.12 | 67.72 | 69.82 | 72.63 | 69.82 |
| WinoGrande | 55.8 | 53.99 | 56.67 | 55.25 | 56.12 | 57.7 | 58.25 | 58.41 |
| Winograd | 71.06 | 68.5 | 67.77 | 70.7 | 73.99 | 70.33 | 72.53 | 72.89 |
| SQuAD v2 (EM) | 29.63 | 35.47 | 28.85 | 34.64 | 36.55 | 34.47 | 39.48 | 36.39 |
| RACE-H | 32.19 | 33.19 | 33.08 | 34.39 | 34.48 | 35.11 | 36.59 | 35.25 |
| RACE-M | 43.8 | 44.22 | 44.22 | 45.96 | 47.7 | 45.75 | 50.07 | 46.59 |
| PIQA | 66.49 | 64.36 | 66.05 | 66.38 | 67.52 | 66.97 | 69.1 | 67.68 |
| ARC-C | 27.99 | 25.77 | 27.65 | 27.22 | 29.01 | 28.75 | 30.55 | 31.48 |
| ARC-E | 56.44 | 54.08 | 58.54 | 57.03 | 59.85 | 58.84 | 63.26 | 61.83 |
| OpenBookQA | 41.4 | 42.2 | 41 | 42 | 43.4 | 42.8 | 44.8 | 45.4 |
| BoolQ | 52.57 | 49.85 | 54.86 | 52.42 | 53.76 | 56.06 | 55.35 | 53.52 |
| COPA | 70 | 75 | 69 | 77 | 74 | 74 | 77 | 75 |
| RTE | 52.35 | 53.07 | 53.07 | 52.35 | 54.15 | 53.43 | 52.35 | 49.82 |
| WiC | 47.34 | 47.34 | 47.18 | 47.34 | 47.34 | 47.34 | 48.43 | 47.02 |
| MultiRC (F1) | 42.98 | 46.69 | 43.82 | 49.09 | 45.29 | 48.2 | 40.99 | 46.42 |
| ReCoRD | 67.32 | 67 | 69.02 | 68.61 | 71.13 | 70.26 | 73.4 | 71.49 |
| CB | 42.86 | 44.64 | 46.43 | 42.86 | 48.21 | 44.64 | 42.86 | 37.5 |
| ANLI-R1 | 32.5 | 33.5 | 31.9 | 33.8 | 33 | 33.3 | 32.4 | 32.1 |
| ANLI-R2 | 30.3 | 34.7 | 30.5 | 34.6 | 30.6 | 33.1 | 31.5 | 33.5 |
| ANLI-R3 | 30.5 | 33.17 | 31.5 | 33.67 | 31.33 | 33.5 | 32.58 | 33.67 |
| Average | 41.96 | 42.71 | 42.84 | 43.85 | 44.85 | 44.51 | 46.21 | 45.13 |
| Avg over GEN Taks | 10.91 | 12.92 | 12.81 | 13.58 | 15.83 | 14.33 | 18.62 | 16.7 |
| Avg over RANK Tasks | 49.35 | 49.8 | 49.99 | 51.06 | 51.76 | 51.69 | 52.77 | 51.9 |
| Dev set log pplx | 2.90 | 2.923 | 2.856 | 2.867 | 2.79 | 2.81 | 2.718 | 2.76 |

Table 15: Downstream Eval numbers and development set log perplexity loss on 2.6B model size granularities.

| Downstream Task | Baseline-S | MatLM-S | Baseline-M | MatLM-M | Baseline-L | MatLM-L | Baseline-XL | MatLM-XL |
|---|---|---|---|---|---|---|---|---|
| TriviaQA (EM) | 18.58 | 18.64 | 19.83 | 21.41 | 25.17 | 24.9 | 28.84 | 28.01 |
| NaturalQuestions (EM) | 3.05 | 3.13 | 3.19 | 3.66 | 4.76 | 4.24 | 6.73 | 5.01 |
| WebQuestions (EM) | 5.61 | 6.74 | 4.43 | 6.3 | 6.1 | 6.74 | 8.27 | 7.78 |
| LAMBADA | 18.46 | 13.74 | 29.92 | 19.89 | 27.34 | 24.84 | 27.94 | 29.98 |
| HellaSwag | 46.41 | 46.01 | 49.04 | 48.94 | 52.87 | 52.2 | 57.14 | 55.33 |
| StoryCloze | 72.26 | 72.1 | 73.54 | 73.22 | 75.09 | 75.04 | 77.02 | 75.79 |
| WSC | 71.23 | 69.82 | 70.88 | 71.58 | 75.09 | 74.39 | 80 | 77.54 |
| WinoGrande | 56.83 | 57.85 | 57.62 | 56.91 | 60.93 | 59.19 | 62.19 | 59.59 |
| Winograd | 76.56 | 71.43 | 72.89 | 74.36 | 76.56 | 74.73 | 81.68 | 78.75 |
| SQuAD v2 (EM) | 34.89 | 37.97 | 34.33 | 40.07 | 34.89 | 42.24 | 43.47 | 42.59 |
| RACE-H | 33.62 | 34.76 | 35.59 | 35.85 | 36.91 | 36.82 | 38.91 | 37.28 |
| RACE-M | 47.63 | 47.49 | 49.44 | 49.51 | 50.77 | 50.07 | 53.34 | 51.67 |
| PIQA | 67.74 | 67.79 | 68.39 | 68.28 | 69.21 | 69.59 | 71.49 | 71.11 |
| ARC-C | 29.95 | 30.29 | 31.83 | 31.91 | 32.51 | 34.22 | 35.67 | 35.41 |
| ARC-E | 60.82 | 59.97 | 61.2 | 62.42 | 63.51 | 64.56 | 67.76 | 64.86 |
| OpenBookQA | 45.6 | 43.8 | 45.4 | 44.8 | 49 | 46.4 | 49 | 49.4 |
| BoolQ | 53.58 | 52.87 | 53.15 | 53.52 | 59.36 | 54.89 | 60.8 | 57.22 |
| COPA | 74 | 74 | 77 | 76 | 75 | 78 | 82 | 81 |
| RTE | 49.1 | 53.07 | 49.82 | 54.15 | 48.01 | 54.51 | 48.01 | 52.35 |
| WiC | 47.34 | 47.34 | 47.18 | 47.34 | 47.34 | 47.18 | 47.02 | 47.49 |
| MultiRC (F1) | 43.4 | 52.28 | 43.65 | 51.64 | 46.99 | 53.7 | 39.24 | 53.77 |
| ReCoRD | 71.34 | 71.9 | 72.79 | 72.97 | 74.86 | 74.57 | 76.71 | 75.32 |
| CB | 28.57 | 44.64 | 46.43 | 46.43 | 41.07 | 50 | 50 | 44.64 |
| ANLI-R1 | 32.4 | 32.3 | 30.4 | 32.3 | 32.5 | 32.1 | 31.2 | 31.5 |
| ANLI-R2 | 30.4 | 30.1 | 30.6 | 31 | 30.1 | 30.2 | 31.7 | 30.8 |
| ANLI-R3 | 30.75 | 30.83 | 31.25 | 31 | 33.5 | 30.92 | 32 | 31.92 |
| Average | 44.23 | 45.03 | 45.76 | 46.36 | 47.29 | 47.93 | 49.54 | 49.08 |
| Avg over GEN Taks | 16.12 | 16.04 | 18.34 | 18.26 | 19.66 | 20.59 | 23.05 | 22.68 |
| Avg over RANK Tasks | 50.93 | 51.94 | 52.29 | 53.05 | 53.86 | 54.44 | 55.85 | 55.37 |
| Dev set log pplx | 2.77 | 2.787 | 2.722 | 2.732 | 2.66 | 2.68 | 2.592 | 2.63 |

Table 16: Downstream eval numbers and development set log perplexity on 2.6B MatLM Mix'n'Match granularities. For original granularities, please refer to Table 15. First row represents the non-embedding parameters of the model.

| Downstream Task | 830M | 1B | 1.11B | 1.32B | 1.43B | 1.55B | 1.65B |
|---|---|---|---|---|---|---|---|
| TriviaQA (EM) | 18.89 | 22.43 | 23.8 | 25.77 | 26.26 | 26.15 | 26.6 |
| NaturalQuestions (EM) | 3.49 | 3.77 | 4.02 | 4.07 | 4.46 | 4.65 | 5.12 |
| WebQuestions (EM) | 5.95 | 6.1 | 6.64 | 6.69 | 6.94 | 6.69 | 6.69 |
| LAMBADA | 16.34 | 20.16 | 23.07 | 24.8 | 24.32 | 25.87 | 29.13 |
| HellaSwag | 47.98 | 50.46 | 51.29 | 52.78 | 53.75 | 54.16 | 54.56 |
| StoryCloze | 73.01 | 73.33 | 74.83 | 75.2 | 75.68 | 75.41 | 75.63 |
| WSC | 70.88 | 70.53 | 74.04 | 72.98 | 74.74 | 73.33 | 77.19 |
| WinoGrande | 57.85 | 58.88 | 60.93 | 58.88 | 59.67 | 60.06 | 59.91 |
| Winograd | 73.26 | 73.26 | 76.19 | 74.36 | 76.56 | 77.66 | 78.02 |
| SQuAD v2 (EM) | 36.49 | 39.72 | 38.05 | 41.33 | 41.08 | 40.26 | 41.36 |
| RACE-H | 34.71 | 35.93 | 35.48 | 36.74 | 36.62 | 36.22 | 36.96 |
| RACE-M | 46.59 | 48.89 | 49.44 | 50.28 | 50.42 | 51.32 | 50.91 |
| PIQA | 68.5 | 69.04 | 69.53 | 70.4 | 70.46 | 70.51 | 70.29 |
| ARC-C | 31.06 | 33.11 | 33.19 | 34.81 | 35.75 | 35.84 | 34.56 |
| ARC-E | 62.29 | 62.58 | 62.63 | 64.86 | 65.99 | 65.49 | 64.69 |
| OpenBookQA | 44.6 | 46.2 | 46.8 | 47 | 47.4 | 47.4 | 47.6 |
| BoolQ | 54.86 | 55.08 | 54.46 | 55.78 | 58.38 | 57.19 | 56.88 |
| COPA | 76 | 76 | 75 | 80 | 77 | 80 | 80 |
| RTE | 53.43 | 53.79 | 53.79 | 52.71 | 53.79 | 54.51 | 53.79 |
| WiC | 47.34 | 47.34 | 47.18 | 47.34 | 47.18 | 47.34 | 48.12 |
| MultiRC (F1) | 53.34 | 53.85 | 52.97 | 54.23 | 57.57 | 55.09 | 54.91 |
| ReCoRD | 72.21 | 73.25 | 73.98 | 74.43 | 74.72 | 75.05 | 75.37 |
| CB | 48.21 | 46.43 | 48.21 | 50 | 50 | 44.64 | 55.36 |
| ANLI-R1 | 32.4 | 32.1 | 32 | 32.4 | 32.3 | 31.4 | 32.4 |
| ANLI-R2 | 30.5 | 30.6 | 30.6 | 30.6 | 30.7 | 30.4 | 31.4 |
| ANLI-R3 | 31.17 | 31.17 | 31.17 | 31.5 | 31 | 31.5 | 31.33 |
| Average | 45.82 | 46.69 | 47.28 | 48.07 | 48.57 | 48.39 | 49.18 |
| Avg over GEN Taks | 16.23 | 18.44 | 19.12 | 20.53 | 20.61 | 20.72 | 21.78 |
| Avg over RANK Tasks | 52.87 | 53.42 | 53.99 | 54.63 | 55.22 | 54.98 | 55.71 |
| Dev set log pplx | 2.774 | 2.729 | 2.706 | 2.68 | 2.675 | 2.663 | 2.65 |