# OpenReview forum: "MatFormer: Nested Transformer for Elastic Inference"
_ICLR.cc/2024/Conference — Submitted to ICLR 2024_

### Official Review · Reviewer_EdnT · 2023-10-30

**Soundness:** 4 excellent
**Presentation:** 4 excellent
**Contribution:** 4 excellent
**Rating:** 8
**Confidence:** 3

**Summary:**

The authors propose techniques for a) training Transformer models such that they have configurable hidden size inside the FFN projection after training and b) mixing/matching hidden sizes across the FFNs in a Transformer model to produce a range of quality / cost tradeoffs for Transformer serving. They evaluate their technique on text and image tasks and demonstrate potential applications like speculative decoding.

**Strengths:**

The paper is well written and organized. I found it easy to follow.

The most compelling part of the paper was the application to speculative decoding, I think. Potential wins from shared parameters, shared attention cache and consistency between the base/full model are very clear.

**Weaknesses:**

There are a number of small things I’d encourage the authors to revise to strengthen their paper.

1) Saying that MatFormer is “zero additional cost” seems inaccurate given the cost of the additional ‘g - 1’ forward passes during training.

2) The authors’ don’t explain how the mix’n’match models on the pareto frontier are selected. Given there are many candidates this seems like an important detail.

3) The claim that the FFN is responsible for the largest chunk of latency during inference seems questionable to me. I’d encourage the authors to present a more nuanced perspective on Transformer inference based on previously published data. For example, the study from Kim et. al [1] presents relevant data for CPU serving in Figures 7 and 8.

4) In the speculative decoding experiments, I’d encourage the authors to briefly describe how consistency between the two models can translate into latency reductions. This will make the impact more clear to readers who aren’t familiar with prior work/do not read the reference you direct them to.

[1] https://arxiv.org/abs/2302.14017

**Questions:**

One ablation I would like to see is how a Transformer performs if the 'g' additional forward passes aren't used during training with your Mix'n'Match procedure. Basically, how bad does a normal Transformer perform with this post-processing if it's not trained appropriately?

---

> ### Author Response · Authors · 2023-11-16
> **Official Response to the Reviewer EdnT**
>
> We thank the reviewer for their support, and their recognition of the potential inference wins. We address your feedback below:
>
> > Saying that MatFormer is “zero additional cost” seems inaccurate given the cost of the additional ‘g - 1’ forward passes during training.
>
> By “zero additional cost”, we mean that the Mix’n’match models are obtained without ever being explicitly optimized for. We will clarify this in the next revision. Moreover, there is no overhead during inference compared to the baseline.
>
> > The authors’ don’t explain how the mix’n’match models on the pareto frontier are selected. Given there are many candidates this seems like an important detail.
>
> Currently, we induce nested substructures in the hidden dimensions of FFN blocks. We tested several heuristics to select the best subnetwork, but consistently observed that gradually using larger granularities in deeper layers worked the best. More formally, we use non-decreasing hidden dimensions with the least slope (change in hidden dimensions across consecutive layers) across layers. This choice behaves nearly optimal (performance lies on the pareto-optimal curve). We will add a discussion on search heuristics used for Mix’n’Match in the next revision.
>
> > The claim that the FFN is responsible for the largest chunk of latency during inference seems questionable to me. I’d encourage the authors to present a more nuanced perspective on Transformer inference based on previously published data. For example, the study from Kim et. al [1] presents relevant data for CPU serving in Figures 7 and 8.
>
> In Kim et. al  [1], Table 4, we note that BERT-Base with 512 sequence length has ~60% of the total FLOPs, whereas attention block takes the remaining. As you recommended, we will refer this paper or other published paper for these numbers in addition to the current ones (Section B and Table 4 in the Appendix)
>
> > In the speculative decoding experiments, I’d encourage the authors to briefly describe how consistency between the two models can translate into latency reductions. This will make the impact more clear to readers who aren’t familiar with prior work/do not read the reference you direct them to.
>
> We agree with your suggestion to add more explanation for speculative decoding and will expand upon our explanation in Section 4.1.1 in the final revision.
>
> > One ablation I would like to see is how a Transformer performs if the 'g' additional forward passes aren't used during training with your Mix'n'Match procedure. Basically, how bad does a normal Transformer perform with this post-processing if it's not trained appropriately?
>
> Applying Mix'n'Match to the baseline performs poorly because vanilla training does not induce nested structure in the network. We will add this result to the Appendix in the next revision.
>
> -----
> We would be happy to discuss any further questions about the work and would appreciate support for acceptance of the paper.

---

> ### Author Response · Authors · 2023-11-20
> **Further questions or concerns?**
>
> We are happy to discuss if anything in the rebuttal needs more clarification or if the reviewer has further questions regarding the paper.

---

> > ### Author Response · Authors · 2023-11-23
> > **Updated Draft in Response to Feedback**
> >
> > We have updated the draft in response to your helpful feedback. In particular, we have:
> >
> > - Explained the Mix'n'Match heuristic in Appendix C.
> > - Added your suggested baseline, which performs poorly if it's not trained appropriately in Figure 7 (Appendix C).
> > - We have added nuance to our discussion of training/inference costs in Appendix B and the Conclusion.

---

### Official Review · Reviewer_J4dY · 2023-10-31

**Soundness:** 2 fair
**Presentation:** 2 fair
**Contribution:** 2 fair
**Rating:** 5
**Confidence:** 4

**Summary:**

The paper presents MatFormer, a model architecture based on the Transformer model. MatFormer employs the concept of matryoshka representation learning to introduce nested sub-structures in feed-forward network (FFN) blocks. The architecture aims to facilitate elastic inference by allowing for the extraction of numerous smaller sub-models without additional training.

**Strengths:**

The paper introduces an interesting idea of elastic inference within the Transformer architecture, a concept that can be potentially beneficial for a wide range of deployment scenarios.

**Weaknesses:**

1. Motivation and Efficiency: While the paper does address the limitations of current large foundation models, it falls short in clearly explaining the training efficiency gains provided by MatFormer. Specifically, there seems to be a lack of substantial evidence to support the claim that training costs are significantly reduced.

2. Similarity to Existing Techniques: The paper does not sufficiently distinguish MatFormer from existing techniques such as mixture-of-experts and other conditional computation methods. This raises questions about the novelty of the work.

3. Scope of Applicability: The focus on FFNs leaves the attention mechanism of the Transformer model unaddressed. This limitation narrows the effectiveness of MatFormer in improving Transformers comprehensively.

4. Evaluation and Support for Claims: The paper could benefit from a more rigorous evaluation to substantiate its claims. As it stands, the contributions asserted lack sufficient empirical validation.

5. Prior Work Comparison: The work could benefit from a clearer discussion of how MatFormer advances beyond or differentiates from key prior work, specifically Kusupati et al., 2022, in terms of the nested structure.

**Questions:**

- The paper seems to use MatLM, MatFormer, and MatViT interchangeably.
- In Figure 2 and the associated text, what are the differences in model architecture / settings among MatFormer, Mix'n'Match, and baseline?
- How does the proposed method improve speculative decoding? Section 4.1.1 is missing certain details to help understand this claim.

---

> ### Author Response · Authors · 2023-11-16
> **Official Response 1/2 to the Reviewer J4dY**
>
> We thank the reviewer for their review, and recognition of how our work can be beneficial for a wide range of deployment scenarios. We address their feedback and questions below:
>
> > Motivation and Efficiency: While the paper does address ... significantly reduced
>
> In Appendix B, we have discussed and empirically demonstrated the training efficiency of MatFormers (our method costs 15% less FLOPs than training independent models). At the same time, unlike the baselines, MatFormer allows extracting 100s of accurate submodels that are not explicitly optimized.
>
> > Similarity to Existing Techniques: The paper does ... the novelty of the work.
>
> Conditional computation methods such as MoE [1] are orthogonal to MatFormer, given that in this work, we co-train Transformer feedforward networks within the same representation space as a pre-training method that is suitable for elastic inference, and do not introduce routing mechanisms. In Table 5 in the Appendix, we have compared our training method with training independent MLP modules, and sampling, which are important features of MoEs - we find that MatFormer performs better than these baselines across all granularities. We will add more discussion on this in our next revision.
>
> > Scope of Applicability: The focus on FFNs ... Transformers comprehensively.
>
> We have chosen to focus on the FFN of the Matformer, given that it forms the bulk of our computational costs (~60%) at the scale we consider (2.6B) and beyond. In this work, we focus on conducting extensive analysis and ablations on this architecture, and plan extending this technique to other components of the Transformer in future work.
>
> > Evaluation and Support for Claims: The paper could benefit from ... asserted lack sufficient empirical validation.
>
> Evaluation and Support for Claims: In our work, we empirically show the effectiveness of the MatFormer architecture across multiple modalities and scales.
> - More specifically, in Section 4.1 we evaluate for decoder-only language modeling on 26 different classification and generation NLP tasks and predict scaling trends across 7 different model scales ranging from 70M to 2.6B.
> - In Section 4.1.1, we conduct experiments that showcase the elasticity of MatFormers, with evaluations on the models obtained using Mix’n’Match showing the linear scaling trend of the extracted submodel, and experiments showing the effectiveness of MatFormer compared to vanilla transformers for speculative decoding.
> - In Section 4.2, we additionally validate our method on image classification and retrieval on ImageNet-1K.
>
> Given our extensive empirical results, we are unsure of which specific claims are unsubstantiated. We are happy to discuss this further and clarify any concerns the reviewer might have.
>
> > Prior Work Comparison: The work could ... key prior work, specifically Kusupati et al., 2022, in terms of the nested structure.
>
> As discussed in related work, MRL (Kusupati et al., 2022) focuses on the nested structure of the output representations. In contrast, MatFormer brings the core idea of nesting from MRL into the model weight/parameter space which helps in flexibility not only in embeddings for large-scale retrieval but also for flexibility in model inference. We mention this in the introduction but will make it clearer in the next revision.
>
> > The paper seems to use MatLM, MatFormer, and MatViT interchangeably.
>
> We use MatLM and MatViT respectively when talking about MatFormers applied to decoder-only language modeling and to vision transformers. We use the more general term, MatFormer, when our exposition applies to both our vision and language modeling experiments. We will carefully check the next revision to ensure that we have used this terminology as intended.
>
> > In Figure 2 and the associated text, what are the differences in model architecture / settings among MatFormer, Mix'n'Match, and baseline?
>
> In Figure 2 the architecture and settings for all three are similar, with the difference between MatFormer and baseline being the training objective. Mix’n’Match models are derived from a trained MatFormer model at inference time by activating different granularities for each layer. We will include these details in the Appendix.

---

> > ### Author Response · Authors · 2023-11-16
> > **Official Response 2/2 to the Reviewer J4dY**
> >
> > > How does the proposed method improve speculative decoding? Section 4.1.1 is missing certain details to help understand this claim.
> >
> > Speculative decoding is an inference time algorithm that combines a smaller model with a larger model to reduce latency while maintaining the accuracy of the larger model. This is done by using the smaller LM to autoregressively generate a few tokens (drafts), and then using the larger LM to verify the accuracy of these drafts. If the draft is inaccurate, the larger LM takes over to correct the prediction. Much of the expense and slow down of this algorithm stems from cases where the smaller model’s predictions disagree with the larger model. In MatFormer, the smaller and larger models agree with each other significantly more often (Figure 2d), resulting in far fewer rollbacks and significantly lower latency as shown in Table 1. Moreover, in our case it is possible to share the attention cache, which results in further latency improvements.  We included a more concise explanation of this in 4.1.1 due to space constraints, and will expand upon this in the updated draft for clarity.
> >
> > [1] Switch Transformers: Scaling to Trillion Parameter Models with Simple and Efficient Sparsity, 2021
> >
> > -----
> >
> > We hope that the rebuttal clarifies the questions raised by the reviewer. We would be  happy to discuss any further questions about the work, and would appreciate an appropriate increase in the score if the reviewer’s concerns are adequately addressed.

---

> ### Author Response · Authors · 2023-11-20
> **Further questions or concerns?**
>
> We are happy to discuss if anything in the rebuttal needs more clarification or if the reviewer has further questions regarding the paper.

---

> > ### Author Response · Authors · 2023-11-23
> > **Updated Draft in Response to Feedback**
> >
> > We have updated the draft in response to your helpful feedback. In particular, we have:
> > - Expanded upon our explanation of SPEED and how MatLM improves SPEED vs baselines in Section 4.
> > - Added details on Mix'n'Match in Appendix C.
> > - Clarified how we differ from MRL  (Kusupati et al., 2022) in Related Work (Section 2)

---

### Official Review · Reviewer_UoXf · 2023-11-01

**Soundness:** 2 fair
**Presentation:** 3 good
**Contribution:** 3 good
**Rating:** 6
**Confidence:** 4

**Summary:**

This paper proposes a nested Transformer architecture called MatFormer based on principle of matryoshka representation learning, designed to offer elasticity in a variety of deployment constraints. During training, each feed forward network (FFN) block of a MatFormer model is jointly optimized with a few nested smaller FFN blocks. And this paper propose to jointly optimize all the submodels together by combining their loss together. During elastic inference, MatFormer allows for the Mix’n’Match of model granularities across layers, i.e., a trained universal MatFormer model enables extraction of hundreds of accurate smaller models. The design of MatFormer can be applied to both decoder and encoder networks.

**Strengths:**

1. The proposed nested Transformer for elastic inference is possible to extract exponentially many submodels based on the deployment constraints rather than only few submodels. I appreciate this high flexibility of MatFormer.

2. The authors fully explore the design and extension of this solution. For example, they explore to reweight different granularities of submodels in Table 6, and evaluate MatFormer spanning from different modalities, i.e., language and vision, model classes, i.e., decoder and encoder, and scales (up to 2.6B parameters) in their main experiments.

3. I think this research is valuable and general for the LLM inference, and has a great potential impact on the design and application deployment of large foundation models in the future.

**Weaknesses:**

1. The parameters g in the article, i.e., logarithmically spaced granularity, is an important parameter for MatFormer. The authors selected g = 4 for experimental verification and analysis. I wonder about the impact of different values of g on the flexibility, training efficiency and effectiveness of MatFormer, and the consistency and accuracy of submodels. This should be an interesting and important exploration which is currently missing.
2. The Mix'n'Match procedure can freely combine hundreds of consistent submodels based on MatFormer layers to meet various specific constraints, but the experiments in Section 4 and Appendix seem to have only 9 submodels evaluated (as shown in Figure 2 and 4). I think the author can give more model loss and consistency data results of different combinations of submodels to better support the above advantage of MatFormer.
3. As mentioned in the first paragraph of the Introduction, many similar practical solutions provide models with 7B, 13B, 33B and 65B parameters, but the experiments of this paper only verify and analyze models with parameters between 0.8-2.6B. If the authors have the enough computing resource, it will be quite valuable for LLM research to evaluate the effectiveness and application potential of MatFormer over a even larger model in your future work. This is not a big concern of this work.
4. As authors say in the Introduction: “MatFormer can also form a similar sub-structure on attention heads”. It will be more clear if authors can illustrate the nested structure of attention heads just like Figure 1.
5. The core method is similar to a previous work NetAug [1], which expands a tiny model into several larger models, and the tiny model obtains extra supervision via additional gradients. However, the starting points of two papers are different. NetAug aims to make tiny models more accurate while MatFormer aims to allow elastic Transformer inference. If authors can explain more comparison/analysis between them, I will appreciate the value of this work even more.
6. Although the experiments are evaluated on wide applications and settings, the baseline seems weak to support the improvement of MatFormer. I recommend authors to add more comparisons with stronger baseline methods including other elastic inference techniques.

[1] Cai H, Gan C, Lin J, et al. Network augmentation for tiny deep learning[J]. arXiv preprint arXiv:2110.08890, 2021.

**Questions:**

All of my concerns are explained in the Weaknesses part above.

---

> ### Author Response · Authors · 2023-11-16
> **Official Response to the Reviewer UoXf**
>
> We thank the reviewer for their response, and for their appreciation of how the flexibility of our design could greatly impact the deployment of large foundation models. We address their concerns below:
>
> > The parameters g … which is currently missing.
>
> Thank you for the interesting comment. We note that increasing $g$ increases the number of accurate subnetworks to select from (since we are explicitly optimizing for more models) at the cost of increased training overhead - the inverse holds for decreasing $g$. We refer the reviewer to Table 29 in the MRL paper (Kusupati et al, 2022) for a comparison of logarithmically nested and uniformly distributed granularities.
>
> > The Mix'n'Match procedure … the above advantage of MatFormer.
>
> Yes, the Mix'n'Match procedure indeed results in more models than the 9 points we have included for clarity in the results. We will include more data points in the Appendix of the next version.
>
> > As mentioned in the first paragraph of the … not a big concern of this work.
>
> Due to resource constraints, we are unfortunately unable to scale our empirical results beyond 2.6B. In lieu of this, we have done extensive scaling law experiments on a range of model sizes (70M-2.6B) to show that our techniques are likely to hold at scale. More specifically, we show that the loss of MatFormer scales as well as that of the baselines in Section 4.1.2, and that the consistency of all MatFormer granularities scales significantly better than that of the baselines in Figure 10.
>
> > As authors say in the Introduction: “MatFormer can also form a similar sub-structure on attention heads”. It will be more clear if authors can illustrate the nested structure of attention heads just like Figure 1.
>
> Background: For a given layer let there be $A$ attention heads, with the i-th head projection as $a_i$, each having dimension $d_p$. In a usual Transformer, we concatenate these projections and then project it to output dimension $d$ using $W\in \mathbb{R}^{A\cdot d_p \times d}$. That is: $output = concat(a_1, \cdots, a_{A}) \cdot W$
>
> Nested substructure in attention heads can be introduced using only the first few attention heads. For example, using first $m$ attention heads would result in $output = concat(a_1,\cdots,a_m) \cdot W[:m\cdot d_p]$. Using different nested values of $m$ for different granularities will induce MatFormer nesting in attention heads. We will include this discussion in our next revision.
>
> > The core method … this work even more.
>
> Thank you for bringing NetAug [1] to our attention. NetAug aims to train a single small target model. The training involves sampling auxiliary networks from a SuperNetwork in each step, where the auxiliary network contains the target model as a subnetwork, and jointly training the auxiliary and the target network. These sampled auxiliary networks are not nested. Moreover, the auxiliary networks are not trained to be accurate. In comparison, MatFormer training gives accurate subnetworks of various sizes. We will add a discussion about NetAug in our next revision.
>
> > Although the experiments are … including other elastic inference techniques
>
> We respectfully disagree with the reviewer that the baselines are weak. The baselines are the corresponding architectures trained from scratch. These will yield the highest performance compared to any other training technique to get a model of the same architecture. We additionally refer the reviewer to Table 5 in the Appendix, where we have compared our method to sampling and training independent MLP modules. Here, we find that our method is the most performant for the same number of tokens.
>
> Inference-time speedup techniques for pretrained networks such as techniques which make inference fast for any pretrained network, like Deja Vu [2], Flash Attention [3], and Speculative Decoding [4], can be applied to our spliced subnetworks as well and are complementary to our contributions.
>
> There are other techniques like distillation that might be stronger baselines, but they create significant training overhead. Moreover, these techniques can also be applied to MatFormer to enhance the quality.
>
> [1] Network augmentation for tiny deep learning, ICLR 2022
>
> [2] Deja Vu: Contextual Sparsity for Efficient LLMs at Inference Time, ICML 2023
>
> [3] FlashAttention: Fast and Memory-Efficient Exact Attention with IO-Awareness, NeurIPS 2022
>
> [4] Fast Inference from Transformers via Speculative Decoding, ICML 2023
>
> ----
>
> We hope that the rebuttal clarifies the questions raised by the reviewer. We would be happy to discuss any further questions about the work, and would appreciate an appropriate increase in the score if the reviewer’s concerns are adequately addressed.

---

> ### Author Response · Authors · 2023-11-20
> **Further questions or concerns?**
>
> We are happy to discuss if anything in the rebuttal needs more clarification or if the reviewer has further questions regarding the paper.

---

> > ### Author Response · Authors · 2023-11-23
> > **Updated Draft in Response to Feedback**
> >
> > We have updated the draft in response to your helpful feedback. In particular, we have:
> > - Added more datapoints for Mix'n'Match in Appendix C.
> > - Clarified how our technique can be extended to attention heads in Section 1.
> > - Reference NetAug in Related Work (Section 2)

---

### Official Review · Reviewer_CZp9 · 2023-11-05

**Soundness:** 3 good
**Presentation:** 3 good
**Contribution:** 2 fair
**Rating:** 5
**Confidence:** 3

**Summary:**

This research introduces MatFormer, a novel nested Transformer architecture that addresses the challenges of deploying models across diverse constraints. It optimizes each FFN block in a way that allows for the flexible use of different model sizes across layers, even when they were not explicitly optimized. MatFormer proves to be effective in various model classes, modalities, and scales, with the ability to extract smaller models from a 2.6 billion parameter decoder-only MatFormer language model. These smaller models maintain comparable performance to independently trained counterparts. Additionally, MatFormer-derived encoders maintain the metric-space structure for large-scale retrieval, and speculative decoding with submodels extracted from MatFormer reduces inference latency. This work offers a promising solution for deploying Transformers in a wide range of settings while maintaining fine-grained control over trade-offs like latency, cost, and accuracy.

**Strengths:**

- This paper is well-written and easy to follow.
- To my best knowledge, the research is the first work to introduce an interesting concept such as a nested Transformer architecture for LLMs. The proposed method aims to address the challenges of deploying models across diverse constraints.
- The proposed method demonstrates the ability to obtain a variety of models without the need for additional training after a single learning process.
- The study encompasses a broad spectrum of experiments, spanning both language and vision domains, and incorporating a range of modalities, classes, and scales.

**Weaknesses:**

The reviewer has two main concerns:

**1. Weak baselines: there is no comparison with efficient LLM techniques that enhance inference speed without fine-tuning or additional training.**
- MatFormer's baseline is limited to a vanilla transformer of the same size. However, it would be meaningful to compare it with recent techniques that improve inference speed without fine-tuning or additional training. For example, approaches like Dejavu [1], which leverages contextual sparsity during the inference phase to achieve enhanced inference speed within a given budget, could be an option. It is necessary to conduct such comparisons with several of these baselines to thoroughly evaluate the performance of MatFormer.

**2. Lack of technical contributions compared to pre-trained supernet (the largest transformer)-based hardware-aware NAS methods.**

Background: In the field of NAS, there are studies that consider the largest supernet in the search space as a nested network of subnetworks and train the supernet to provide pre-trained subnetworks optimized for inference budgets [2, 3]. The definition of a search space may vary, but it typically includes various architectural design choices such as the number of blocks, layers, and hidden dimensions of layers.

While the proposed MatFormer focuses on pre-trained LLMs, hardware-aware NAS methods [2] based on pre-trained transformers focus on traditional transformers. Therefore, their settings are not exactly the same. Nevertheless, both methods are fundamentally grounded in transformer structures, with the shared objective of constructing, training elastic transformer-based models, and delivering pre-trained transformer-based models that are optimized for specific inference budget constraints. Therefore, the reviewer finds it meaningful to compare the two in terms of their technical contributions.

- Lack of search algorithm: MatFormer leaves the search algorithm as future work and employs a rather naive method called "Mix’n’ Match" to select the final optimal model within a given inference budget. However, as well-known in NAS research, searching for an optimal model from a search space significantly impacts the final performance and is not a trivial problem. The reviewer thinks that MatFormer would be better off including an algorithm to search for the best combination among pre-trained blocks.

- Simple search space: The proposed MatFormer defines a search space by only considering the number of FFN blocks as the architectural design choice (If I am wrong, please let me know). This seems much simpler than the search spaces designed by NAS methods (e.g., [2]).


[1] Deja Vu: Contextual Sparsity for Efficient LLMs at Inference Time, ICML 2023

[2] HAT: Hardware-Aware Transformers for Efficient Natural Language Processing, ACL 2020

[3] Once-for-All: Train One Network and Specialize it for Efficient Deployment, ICLR 2020

**Questions:**

The reviewer believes that the proposed approach is valuable because it introduces elastic LLMs that provide multiple pre-trained transformer-based models optimized for specific inference budget constraints with a single training for the first time. However, its technical contributions are limited, and the baseline models used for comparison are weak.

- Please address the concerns in Weaknesses section.
- Q. How do training time and memory usage change when using the proposed approach for model training compared to training a single model?

---

> ### Author Response · Authors · 2023-11-16
> **Official Response 1/2 to the Reviewer CZp9**
>
> We thank the reviewer for their comments and suggestions. We are glad that the reviewer found the paper well-written, the idea to be novel and effective for the challenges at hand, and the experiments to be extensive. We address the concerns raised by the reviewer below:
>
> > Weak baselines … the performance of MatFormer
>
> Thank you for highlighting Deja Vu [1] which is a complementary approach to MatFormer. MatFormer’s main focus is designing transformer blocks that have nested structures so that smaller models can be extracted without any further training. In contrast, the goal of Deja Vu is to route tokens through different parts of the model based on sparsity. Deja Vu can be applied to any pretrained network, including spliced subnetworks of MatFormer, to speed up their inference.
>
> > Lack of technical … of their technical contributions.
>
> Thank you for bringing HAT [2] to our attention. The contributions of work like Once-for-all [3] and HAT differ in numerous ways compared to MatFormer:
>
> - HAT vs MatFormer: HAT samples random (not nested) subnetworks during training at each step to optimize. During inference, it conducts NAS to find the subnetwork architecture and then trains it from scratch before deployment. In contrast, MatFormer trains one model and reads off accurate nested models without any further training.
> - Once-for-all (OVA) vs MatFormer: OVA first trains a teacher model, and fine-tunes subnetworks using knowledge distillation. In contrast, MatFormer doesn’t require any finetuning and maintains just a single model.
>
> In addition, both OVA and HAT focus on smaller-scale end devices. OVA trains models of size <=10M params, while HAT trains <=60M param models. In comparison, we train models at a much larger scale (up to 2.6B parameters). We will add the comparison with HAT in the next revision.
>
> > Lack of search algorithm: MatFormer … best combination among pre-trained blocks
>
> We experimented with several heuristics to select the best subnetwork, but consistently observed that gradually using larger granularities in deeper layers worked the best. More formally, we use non-decreasing hidden dimensions with the least slope (change in hidden dimensions across consecutive layers) across layers. Given that this choice behaves nearly optimally (performance lies on the pareto-optimal curve), we did not focus on search techniques. In future work, we plan to extend the nested substructure to other components of the Transformer like attention heads, model dimensions, and n(layers), and will explore NAS methods then. We will add a discussion on search heuristics used for Mix’n’Match in the next revision.
>
> > Simple search … designed by NAS methods
>
> We would like to clarify that we do not choose among the number of FFN blocks, but choose the hidden dimension in each FFN block. Hence, simple search techniques like MixnMatch work well, unlike existing works that require careful NAS. In the future when we extend our work to induce elasticity in other components of the Transformer, we will explore more complex techniques like NAS.
>
> > The reviewer … used for comparison are weak
>
> We respectfully disagree with the reviewer that the baselines are weak. The baselines are the corresponding architectures trained from scratch. These will yield the highest performance compared to any other training technique to get a model of the same architecture. We additionally refer the reviewer to Table 5 in the Appendix, where we have compared our method to sampling and training independent MLP modules. Here, we find that our method is the most performant for the same number of tokens.
> Inference-time speedup techniques for pretrained networks such as techniques which make inference fast for any pretrained network, like Deja Vu [1], Flash Attention [4], and Speculative Decoding [5], can be applied to our spliced subnetworks as well and are complementary to our contributions.
>
> There are other techniques like distillation that might be stronger baselines, but they create significant training overhead. Moreover, these techniques can also be applied to MatFormer to enhance the quality.

---

> > ### Author Response · Authors · 2023-11-16
> > **Official Response 2/2 to the Reviewer CZp9**
> >
> > > How do training time and memory usage change when using the proposed approach for model training compared to training a single model?
> >
> > - **Training FLOPS/Wall Clock Time**: Compared to training the $g$ subnetworks of $g$ granularities independently from scratch, the proposed method saves a modest amount of FLOPs and training time (15% reduction) because of joint optimization. We refer the reviewer to Appendix B and Table 4 for more details. Moreover, our method allows the extraction of 100s of submodels of varying size that were never explicitly optimized for, saving training costs for these 100s of models.
> > - **Memory**: For training, the peak memory usage is roughly equal to the sum of memory usage for the independently trained baselines. On the other hand, at inference time, both baseline and MatFormer have the same memory footprint.
> >
> > [1] Deja Vu: Contextual Sparsity for Efficient LLMs at Inference Time, ICML 2023
> >
> > [2] HAT: Hardware-Aware Transformers for Efficient Natural Language Processing, ACL 2020
> >
> > [3] Once-for-All: Train One Network and Specialize it for Efficient Deployment, ICLR 2020
> >
> > [4] FlashAttention: Fast and Memory-Efficient Exact Attention with IO-Awareness, NeurIPS 2022
> >
> > [5] Fast Inference from Transformers via Speculative Decoding, ICML 2023
> >
> > ------
> > We would be happy to discuss any further questions about the work, and would appreciate an appropriate increase in the score if the reviewer’s concerns are adequately addressed.

---

> ### Author Response · Authors · 2023-11-20
> **Further questions or concerns?**
>
> We are happy to discuss if anything in the rebuttal needs more clarification or if the reviewer has further questions regarding the paper.

---

> ### Author Response · Authors · 2023-11-23
> **Updated Draft in Response to Feedback**
>
> We have updated the draft in response to your helpful feedback. In particular, we have:
> - Clarified your questions on training cost in Appendix B
> - Added details on Mix'n'Match in Appendix C.
> - Discussed HAT in Related Work (Section 2)

---

> ### Comment · Reviewer_CZp9 · 2023-11-23
> **Response by the reviewer CZp9**
>
> Thank you to the authors for their response. I have carefully reviewed both the authors' reply and the comments from other reviewers.
>
> It seems that the reviewers share common concerns, namely: 1) the lack of empirical evidence to support that Mix'n'Match selected pareto-optimal or necessity to improve Mix'n'Match algorithm. 2) weak baselines 3) suboptimal search space - dimension of FFN blocks.  However, I did not find any indication in the authors' response that they suggested improved algorithms or the definition of a new search space or enough empiricial evidence. These aspects appear to be the main limitations that need to be addressed.
>
>  Therefore, my concerns are still remained and I am inclined to maintain the original score.

---

### Author Response · Authors · 2023-11-23
**Updated Draft in Response to Feedback**

We thank the reviewers for their recognition of how the flexibility of our design could greatly impact the deployment of large foundation models in multiple scenarios and their potential to further improve the performance of inference time algorithms like speculative decoding. We have updated the draft in response to their helpful feedback (text highlighted in blue), and would appreciate an appropriate increase in score if these improvements to the draft and additional experiments clarify all reviewer concerns. More specifically:

- We have included an extended discussion of the Mix’n’Match algorithm in Appendix C.
- In Appendix C we have also included extended results on more possible subnetworks in addition to those shown in Figure 2, and the baseline suggested by Reviewer EdnT.
- We have expanded our explanation of speculative decoding (Section 4), and added a discussion on extending our technique to be used in attention heads (Section 1).
- In the Related Work (Section 2), we added additional references to HAT and NetAug, and further clarified how we differ from MRL.
- We have added nuance to our discussion of inference costs in Appendix B and the Conclusion as suggested by the reviewers.

We would be happy to discuss any further questions and clarifications about the work.

---

### Meta-Review · Area_Chair_rafS · 2023-12-09

**Metareview:**

This paper proposes a nested transformer architecture for offering compute elasticity in various deployment constraints. The initial reviews were mixed. On the positive side, the reviewers acknowledged the problem's importance, the writing's quality, and a potentially wide application of the method. However, they also raised concerns regarding the novelty of the approach compared to Once-for-all, supernet, and NAS-based approaches in general. There are also criticisms around the limited scope of applying nested computation to the FFN blocks (ignoring attention), missing details on how final pareto-front architectures are constructed, and missing NAS-based baselines and in-depth comparison and discussion. Given these concerns the submission does not seem to be ready for presentation at ICLR.

**Justification For Why Not Higher Score:**

The submission require further improvements to better contextualize its merits compared to the prior arts.

**Justification For Why Not Lower Score:**

N/A

---

### Decision · Program_Chairs · 2024-01-16

Reject